**TOOLS**

_JCB_ Journal of Cell Biology

# Stage- and tissue-specific gene editing using 4-OHT–inducible Cas9 in whole organism

Yaqi Li[1,2]*, Weiying Zhang[1,3]*, Zihang Wei[1]*, Han Li[1], Xin Liu[1], Tao Zheng[1], Tursunjan Aziz[1], Cencan Xing[4], Anming Meng[1,5], and Xiaotong Wu[1]

Vertebrate genes function in specific tissues and stages, so their functional studies require conditional knockout or editing. In zebrafish, spatiotemporally inducible genome editing, particularly during early embryogenesis, remains challenging. Here, we establish inducible Cas9-based editing in defined cell types and stages. The nCas9[ERT2] fusion protein, consisting of Cas9 and an estrogen receptor flanked by two nuclear localization signals, is usually located in the cytoplasm and efficiently translocated into nuclei upon 4-hydroxytamoxifen (4-OHT) treatment in cultured cells or embryos. As a proof of concept, we demonstrate that genes in primordial germ cells in embryos and germ cells in adult ovaries from a transgenic line with stable expression of nCas9[ERT2] and gRNAs can be mutated by 4-OHT induction. The system also works in early mouse embryos. Thus, this inducible nCas9[ERT2] approach enables temporospatial gene editing at the organismal level, expanding the tissue- and stage-specific gene-editing toolkit.

## Introduction

The boom in genome-editing technologies in recent years, specifically those using the CRISPR/CRISPR-associated (Cas) system, has helped simplify the process of generating genetically engineered animal models (Cho et al., 2013; Cong et al., 2013; Hwang et al., 2013; Li et al., 2013; Wang et al., 2013; Yang et al., 2014). However, traditional whole-body knockout in early embryos via zygotic injection of Cas9 and gRNAs makes it inapplicable to elucidate cell type– and developmental stage-specific functions of genes. Thus, inducible conditional knockout technologies are greatly desired in these circumstances. Many efforts have been made previously to improve traditional CRISPR/Cas9 technology to achieve cell- or tissue-specific genome editing at the organismal level by driving Cas9 expression under a tissue-specific promoter (Ablain et al., 2015; Ablain et al., 2022; Bai et al., 2016; Carroll et al., 2016; Li et al., 2023; Poe et al., 2019) or using a system that works only in particular biological contexts. For example, combined with the Gal4/upstream activation sequence (UAS) binary system (Delventhal et al., 2019; Di Donato et al., 2016; Isiaku et al., 2021; Meltzer et al., 2019; Port et al., 2020) or the Cre/loxP system (Hans et al., 2021; Oldrini et al., 2018; Platt et al., 2014), Cas9 expression could be controlled by the tissue-specific Gal4 transcription factor or the Cre recombinase, in which UAS and the Stop cassette have been placed

upstream of the Cas9 coding sequence, respectively. On the other hand, temporally controlled genome-editing technologies have been applied more to cultured cells. For example, Cas9 expression can be repressed by the tet repressor (TetR) protein via a tetracycline operator (TetO) and activated when doxycycline (DOX) treatment blocks TetR (Katigbak et al., 2018). In addition, assembling of the split Cas9 protein fragments induced by small molecules or light (Nihongaki et al., 2015; Zetsche et al., 2015) or alteration of the Cas9 protein conformation triggered by small molecules (Davis et al., 2015; Oakes et al., 2016) have also been applied to achieve temporal control of genome editing. However, due to uncommon specialized equipment, the complexity of the expression systems, or limited cell permeability of inducible drugs, only a few of these inducible CRISPR/Cas9 systems have been utilized _in vivo_ (Katigbak et al., 2018; Senturk et al., 2017).

In mice, inducible tissue-specific genome-editing technologies based on Cre recombinase were widely used, such as TetO-Cre induced by DOX (Jensen et al., 2022; Liu et al., 2014b; Miao et al., 2021; Zhang et al., 2013) and the Cre variant with estrogen receptor induced by the xenoestrogen tamoxifen (Hayashi and McMahon, 2002; Indra et al., 1999; Southard et al., 2014; Xu et al., 2017). The subcellular localization and activity of the Cre recombinase can be regulated by tamoxifen or a related analog,

[1]Laboratory of Molecular Developmental Biology, State Key Laboratory of Membrane Biology, Tsinghua-Peking Center for Life Sciences, School of Life Sciences, Tsinghua University, Beijing, China;   [2]Department of Histology and Embryology, Basic Medical College, Naval Medical University, Shanghai, China;   [3]Department of Obstetrics and Gynecology, Seventh Medical Center of Chinese PLA General Hospital, Beijing, China;   [4]School of Chemistry and Biological Engineering, University of Science and Technology Beijing, Beijing, China;   [5]Guangzhou Laboratory, Guangzhou, China.

*Y. Li, W. Zhang, and Z. Wei contributed equally to this paper.   Correspondence to Xiaotong Wu: wuxt@mail.tsinghua.edu.cn;   Anming Meng: mengam@mail.tsinghua.edu.cn.

such as 4-hydroxytamoxifen (4-OHT), when Cre is linked with a mutated ligand-binding domain of the estrogen receptor (ERT2) to avoid endogenous estrogen interference (Indra et al., 1999; Metzger et al., 1995). Notably, the Cre-based system requires the knock-in of two loxP fragments into the target gene locus (Antonson et al., 2012; Hall et al., 2009; Sauer and Henderson, 1989). Therefore, the popularity of the inducible Cre/loxP system in zebrafish is largely impeded by the low knock-in germline transmission rate in zebrafish (Albadri et al., 2017; Kesavan et al., 2018; Mi and Andersson, 2023; Prykhozhij et al., 2018). Moreover, fusion of ERT2 to Cas9 triggers nuclear translocation of Cas9 protein upon 4-OHT treatment, which has been demonstrated in mammalian cell lines (Liu et al., 2016; Zhao et al., 2018). Thus, we speculated that Cas9-ERT2 might also render highly efficient and convenient spatiotemporal gene editing in zebrafish.

Early development of zebrafish embryos is regulated by many maternally expressed genes (Fuentes et al., 2020; Langdon and Mullins, 2011), some of which are also expressed zygotically. It has been observed that zygotic mutants of some maternal/zygotic genes are embryonic lethal, making it difficult to obtain their maternal mutants for investigating functions of their maternal products (Ciruna et al., 2002). Regarding germ cells (GCs), their development involves interaction with surrounding somatic cells (SCs), in which a single gene may be expressed in both cell types; thus, elucidation of its function in a specific cell type is required. In this study, therefore, we made an effort to establish an inducible gene knockout system, based on CRISPR/Cas9 technology, to achieve temporal gene knockout in zebrafish GCs.

## Results

### Design and test of 4-OHT–inducible nucleus-localizing Cas9

To establish a 4-OHT–inducible Cas9 system, we made several pCS2 vector-based expression constructs using zebrafish codon-optimized Cas9 (Liu et al., 2014a), ERT2, SV40 nuclear localization signal (NLS1), and nucleoplasmin NLS (NLS2) (Fig. S1 A). After transfection into HEK293T cells, we found that only Cas9-NLS1-ERT2-NLS2 protein (nCas9$^{ERT2}$) showed nuclear transport upon 5 μM 4-OHT treatment (Fig. 1 A and Fig. S1 B), suggesting the importance of NLS positioning in the fusion protein for its inducible nuclear translocation. To test the ability of nCas9$^{ERT2}$ to edit target genes, its expression vector and the plasmid pU6a-EMX1-gRNA for targeting the human EMX1 locus (Cong et al., 2013) were co-transfected into HEK293T cells. T7 endonuclease I (T7EI) assay revealed that the EMX1 locus was successfully edited only after 4-OHT treatment, which was similar to the editing effect of expressing nCas9 and EMX1 gRNA from a single plasmid (Fig. 1 B). Therefore, nCas9$^{ERT2}$ was used in subsequent studies.

### 4-OHT–induced nuclear translocation of nCas9$^{ERT2}$ in zebrafish primordial GCs

The zebrafish piwil1 (also called ziwi) has been reported to be expressed in primordial GCs (PGCs) in early embryos and in oogonia and oocytes in adults (Houwing et al., 2007; Houwing et al., 2008). Based on the previously reported 4.8-kb (kb pairs) piwil1 promoter with germline transcriptional specificity (Leu and Draper, 2010), promoter dissection by transient transgenesis in zebrafish identified its 2.5-kb proximal region with transgenic activity in PGCs/GCs (Fig. S2 A). This 2.5-kb piwil1 promoter was used subsequently. To enhance the GC expression specificity, as the 3′UTR sequence of nanos3 (previously named nanos1) mRNA is crucial for PGC location (Doitsidou et al., 2002; Köprunner et al., 2001; Saito et al., 2006; Wong and Collodi, 2013), we made the Tol2 transposon–based construct Tol2 (piwil1:mCherry-CAAX-n3U) by combining the 2.5-kb piwil1 promoter with the nanos3 3′UTR (n3U) to enhance GC-specific localization of expressed target mRNA. In resulting Tg(piwil1:mCherry-CAAX-n3U) germline transgenic embryos, the mCherry fluorescence, as observed by fluorescent microscopy, was clearly restricted to PGCs/GCs from 6 h postfertilization (hpf) (shield stage) to 4 days postfertilization (dpf) (Fig. 1 C). However, immunofluorescence using these embryos detected an mCherry signal in all cells at the 512-cell stage (2.75 hpf), mainly in PGCs at the sphere stage (4 hpf), and then exclusively in PGCs thereafter (Fig. S2 B).

We utilized the 2.5-kb piwil1 promoter, nCas9$^{ERT2}$, and n3U to generate the zebrafish Tg(piwil1:nCas9n$^{ERT2}$-n3U) transgenic line. Immunofluorescence observation indicated that, in Tg(piwil1:nCas9n$^{ERT2}$-n3U) transgenic embryos, Cas9 protein was ubiquitously distributed at the 512-cell stage and became localized in Ddx4/Vasa-positive PGCs at and after the sphere stage (Fig. 1 D). The analysis of the ratio of Cas9-positive PGCs to total PGCs revealed that more than 70% of PGCs expressed Cas9 from the sphere to 24 hpf (Fig. 1 E), which indicates that the optimized 2.5-kb piwil1 promoter and nanos3 3′UTR are sufficient to drive the PGC-specific expression of nCas9$^{ERT2}$. Importantly, nCas9$^{ERT2}$ protein was predominantly present in the cytoplasm. Then, we explored the way to induce nuclear transport of nCas9$^{ERT2}$ in Tg(piwil1:nCas9n$^{ERT2}$-n3U) embryos. These embryos were incubated in 20 μM 4-OHT at normal raising buffer and temperature, kept out of the light from 4 hpf onward, and collected at different time points in an interval of 2 h for Cas9 immunofluorescence (Fig. 2 A). Results indicated that nCas9$^{ERT2}$ protein remained in the cytoplasm of PGCs at 1 h posttreatment (hpt), became detectable in the nucleus at 2 hpt, and was highly enriched in the nucleus at 4 and 6 hpt (Fig. 2 B). Analysis of the Cas9 signal intensity ratio (nucleus to whole cell) revealed that, starting from the shield stage, 4-OHT–treated embryos exhibited significantly higher nucleus-to-whole-cell ratios of nCas9$^{ERT2}$ protein in PGCs compared with control embryos (Fig. 2 C). Thus, nuclear transport of nCas9$^{ERT2}$ can be achieved in PGCs of living embryos by a relatively shorter period of 4-OHT treatment. Besides, we did not see an adverse effect of 4-OHT on embryonic development in our treating conditions.

### Stage-specific editing of tbxta gene in embryos by the inducible nCas9$^{ERT2}$ system

To test whether 4-OHT is able to induce a sufficient amount of nuclear nCas9$^{ERT2}$ to execute stage-specific gene editing in Tg(piwil1:nCas9$^{ERT2}$-n3U) transgenic embryos, we chose to target the tbxta/ntl gene, which is zygotically expressed in mesendodermal

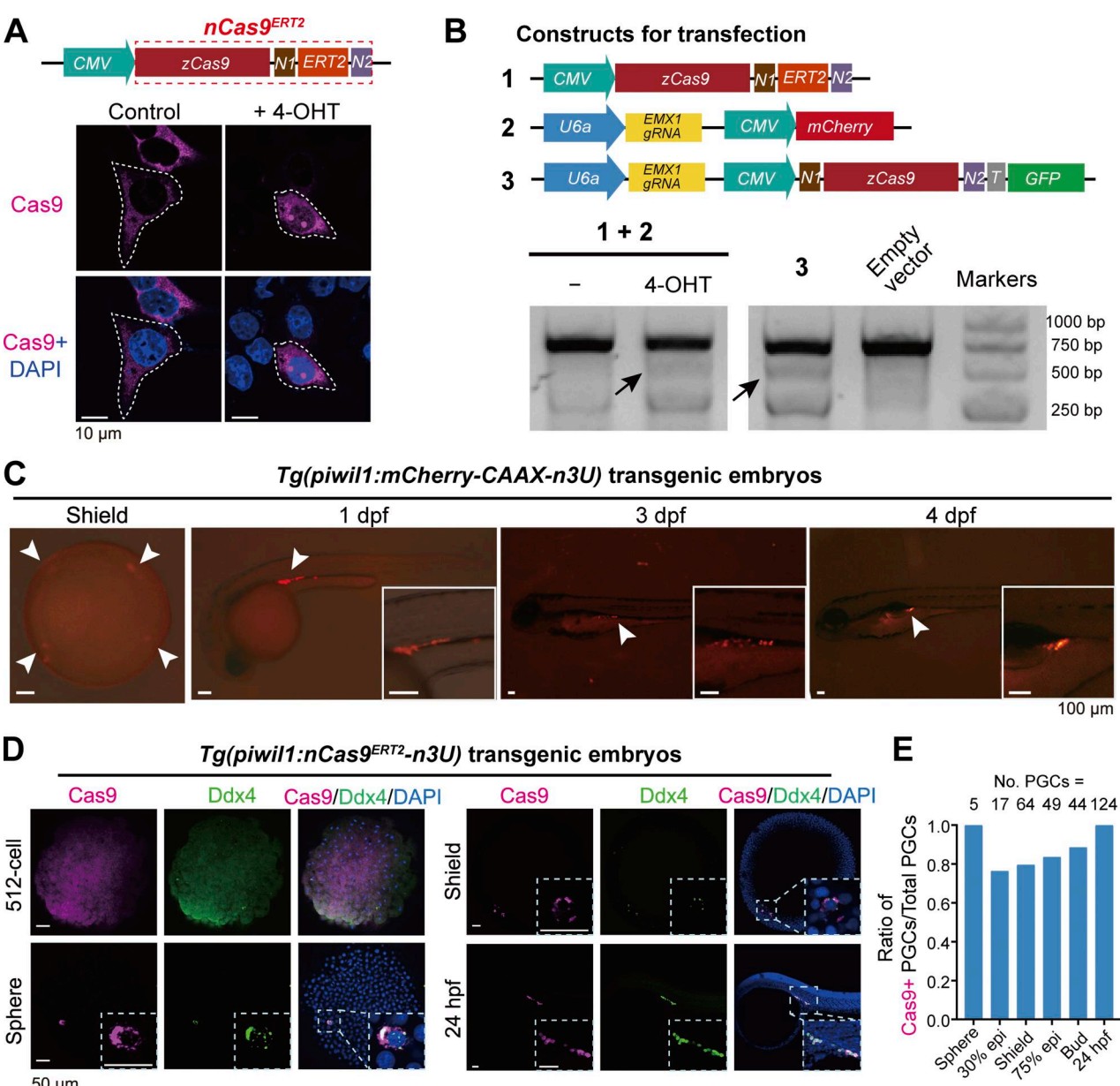

**Figure 1. Nuclear translocation and gene editing effectiveness of nCas9ERT2. (A)** 4-OHT–induced nuclear translocation of nCas9$^{ERT2}$ in human HEK293T cells. Cells were transfected with an expression construct (top) and immunostained with Cas9 antibody and DAPI staining (bottom). The images shown were representative images with Cas9-positive cells outlined by the dashed line. N1, SV40 NLS; N2, nucleoplasmin NLS. **(B)** Gene editing effectiveness of 4-OHT–induced nCas9$^{ERT2}$ in HEK293T cells. Endogenous *EMX1* was the target. Top, compositions of constructs with designated numbers for transfection (T, T2A); bottom, gel electrophoretic image showing T7E1 assay outcome of *EMX1* locus. The mutant band was indicated by arrows. **(C)** The indicated transgenic embryos express mCherry specifically in PGCs (indicated by arrowheads), confirming the PGC specificity of the zebrafish *piwil1* promoter. Inserts showed arrowhead-pointed areas. **(D)** Expression pattern of nCas9$^{ERT2}$ in indicated transgenic embryos at different stages. Embryos were immunostained with Cas9 and Ddx4 (a PGC-specific marker) antibodies plus DAPI staining. Note that Cas9 started to be specifically expressed in PGCs from the sphere stage onward. Inserts showed enlarged areas harboring PGCs. **(E)** Bar plot showing the ratios of Cas9-positive PGCs (Ddx4 and Cas9 double-positive cells) to total PGCs (Ddx4-positive cells) in *Tg(piwil1:nCas9$^{ERT2}$-n3U)* embryos we analyzed in at least three replicates. No. PGCs, the number of PGCs at each stage. Source data are available for this figure: SourceData F1.

progenitors and the notochord (Schulte-Merker et al., 1992). We adopted two strategies: (1) *Tg(piwil1:nCas9$^{ERT2}$-n3U;piwil1: mCherry-CAAX-n3U)* transgenic embryos injected with *tbxta-*gRNA at the one-cell stage and treated with 40 μM 4-OHT from the sphere stage onward (Fig. 3 A, left). (2) Injection of synthetic *tbxta*-gRNA into the one-cell stage *Tg(piwil1:nCas9-n3U;piwil1:mCherry-CAAX-n3U)* transgenic embryos, in which

nCas9 was not fused to ERT2 and thus could transport to the nucleus without 4-OHT induction (Fig. 3 A, right); The resulting embryos were observed at 24 hpf, followed by identification of mutated alleles of *tbxta* using next-generation sequencing (NGS) on isolated mCherry-positive PGCs and mCherry-negative SCs. In the case of *Tg(piwil1:nCas9-n3U)* embryos injected by *tbxta*-gRNA (Fig. 3 A, right, green), 28.9% (30/104)

Figure 2. **4-OHT–induced nuclear translocation of PGC-specific nCas9^ERT2 in transgenic zebrafish embryos. (A)** Schematics showing the mating strategy and 4-OHT treatment of embryos. **(B)** Immunostaining results for Cas9 (magenta) and Ddx4 (green) with nuclei stained by DAPI before and after 4-OHT treatment. The dashed boxes in the first row were enlarged and shown in the three lower rows. **(C)** Box plots showing the ratio of Cas9 (magenta) signal intensity in the nucleus to total PGCs (mean ± SD). Nu, nucleus. Each independent data point represents a PGC. The numbers of PGCs in control groups at 30% epi, shield, 75% epi, and bud stages are 10, 6, 4, and 10, respectively, while the numbers of PGCs in 4-OHT–treated groups at each stage are 4, 5, 11, and 10. P values of the *t* test (unpaired, two-sided) were also labeled. Source data are available for this figure: SourceData F2.

exhibited truncation of the posterior trunk and lack of the notochord at 24 hpf (class III), which phenocopied *tbxta* mutants (Amacher et al., 2002; Marlow et al., 2004), 64.4% (67/104) had a mildly truncated posterior trunk (class II), and the remaining had a normal trunk (class I) (Fig. 3 B). In the case of *Tg(piwil1:nCas9^ERT2-n3U)* embryos treated with the 4-OHT or DMSO at the sphere stage, all (*n* = 86 and 90, respectively) embryos showed a normal notochord morphology (class I) at 24 hpf (Fig. 3 B).

Cloning and sequencing of the *tbxta-gRNA*–targeted region indicated that 70.6% (12/17) and 50.0% (10/20) of alleles were mutated in mCherry-positive PGCs and mCherry-negative SCs in class III embryos at 24 hpf (Fig. 3 C, right), respectively. However, in *Tg(piwil1:nCas9^ERT2-n3U)* embryos treated with 4-OHT, 40.0% of *tbxta* alleles were mutated in PGCs, whereas no mutated alleles were found in SCs or in DMSO-treated PGCs and SCs (Fig. 3 C, left). To further validate no mutations in SCs, we performed whole-genome sequencing (WGS) of PGCs and

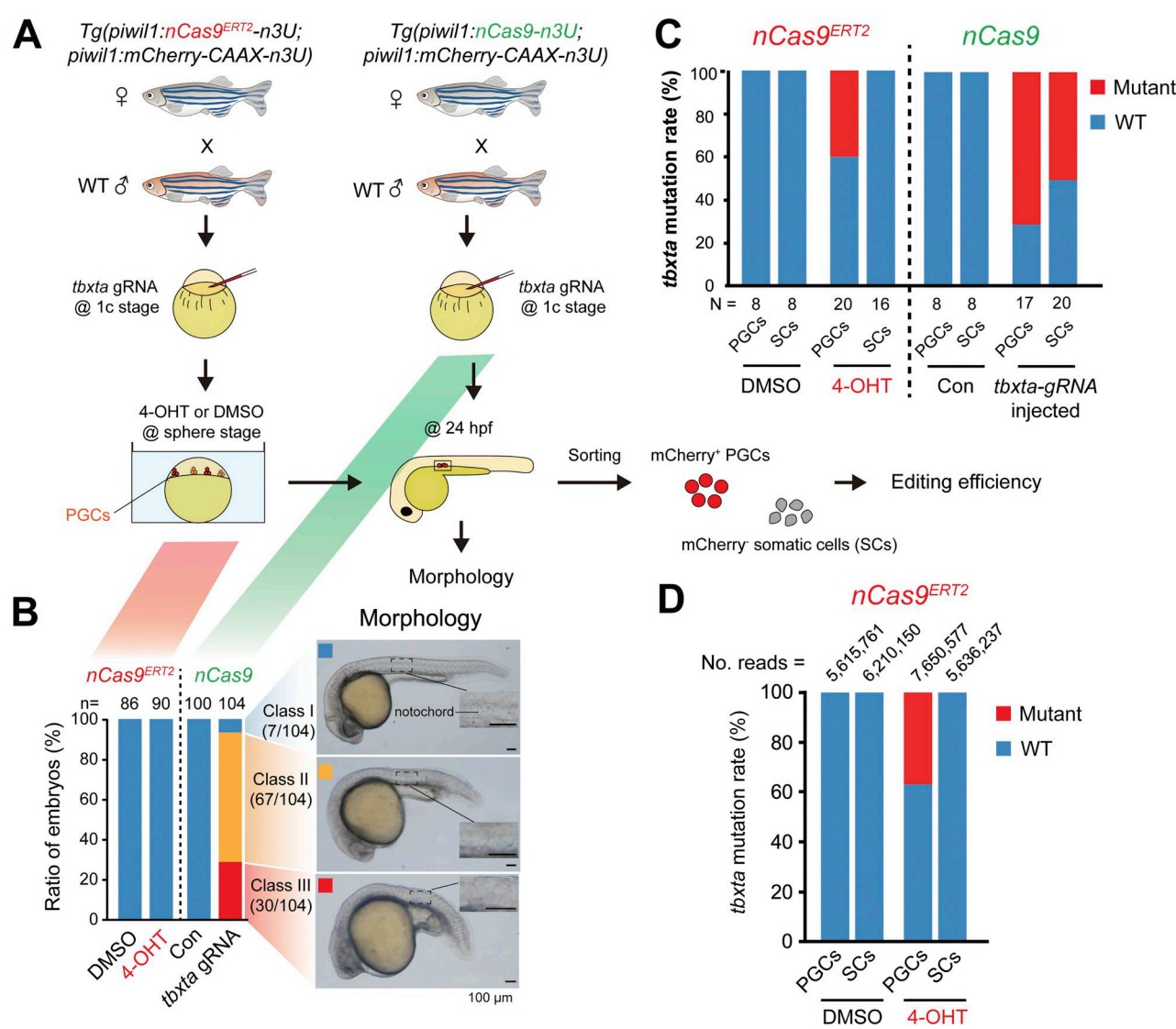

Figure 3. **Specificity and editing efficiency of the nCas9^ERT2 system. (A)** Schematics of experimental procedures. *Tg(piwil1:nCas9^ERT2-n3U)* (left) or *Tg(piwil1: nCas9-n3U)* (right) transgenic females were crossed to WT males, which produced embryos with maternal expression of inducible nCas9^ERT2 or non-inducible nCas9, respectively. **(B)** Left, bar chart showing the ratio of different morphological types of *tbxta*-gRNA–injected 4-OHT– or DMSO-treated nCas9^ERT2 embryos and nCas9 embryos at 24 hpf. *n*, the number of embryos. Right, morphology of representative embryos in each class. The boxed areas of notochord were enlarged. **(C)** Mutation rate of the *tbxta* locus in mCherry-positive PGCs or mCherry-negative SCs in different groups at 24 hpf. A region in the *tbxta* locus was amplified and cloned, followed by sequencing individual clones. *N*, number of sequenced clones. **(D)** Bar chart showing the mutation rate of the *tbxta* locus in mCherry-positive PGCs or mCherry-negative SCs in *tbxta*-gRNA–injected 4-OHT– or DMSO-treated nCas9^ERT2 embryos at 24 hpf, calculated with NGS results. No. reads, the number of aligned reads per target site. Source data are available for this figure: SourceData F3.

SCs from 4-OHT- or DMSO-treated *Tg(piwil1:nCas9^ERT2-n3U)* embryos. The NGS results provided at least 5 million reads for each sample, which is sufficient to determine whether a mutation is present. The results revealed an efficient mutation rate (37.1%) of the *tbxta* target site induced by nCas9^ERT2 in PGCs within 4-OHT–treated embryos (Fig. 3 D), which was comparable with Sanger sequencing results (Fig. 3 C). Whereas there were no mutated reads in PGCs and SCs from the DMSO-treated embryos and SCs from the 4-OHT–treated embryos (Fig. 3 D). In sum, these results indicated that the gene-editing activity of nCas9^ERT2 can be temporally controlled during embryogenesis through the application of 4-OHT.

To accurately assess the mutation efficiency of the *tbxta* locus and further determine whether off-target gene editing exists within this system, we performed WGS with PGCs of 4-OHT– or DMSO-treated *Tg(piwil1:nCas9^ERT2-n3U)* embryos (Fig. S3 A) and found that the mutation rate of the *tbxta* locus in PGCs of 4-OHT– treated embryos is about 27.8%, with no off-target editing in either 4-OHT– or DMSO-treated embryos at predicted potential off-target sites by Cas-OFFinder (Bae et al., 2014) (Fig. S3 B).

### GC-specific editing of *huluwa* gene in young fish by the inducible nCas9^ERT2 system

To test the utility of the inducible nCas9^ERT2 system in adults, we chose to target the *huluwa* (*hwa*) gene, which is a maternal gene,

and its maternal mutants lack the dorsal organizer and body axis (Yan et al., 2018). To this end, we generated the *Tg(U6:hwa-gRNAs;ef1a:GFP)* transgenic line, which simultaneously expresses two previously reported *huluwa* gRNAs (Yan et al., 2018) as well as GFP for easy identification of transgenic fish (Fig. 4 A). By crossing *Tg(piwil1:nCas9$^{ERT2}$-n3U)* fish with *Tg(U6:hwa-gRNAs;ef1a:GFP)* fish, we obtained *Tg(piwil1:nCas9$^{ERT2}$-n3U;U6:hwa-gRNAs;ef1a:GFP)* double transgenic zebrafish. When grown to 45 dpf, these fish were treated twice, each lasting 12 h with an interval of 12 h for resting, with 5 μM 4-OHT, followed by raising in normal conditions and keeping out of the light. After reaching sexual maturity (about 3 mo postfertilization (mpf)), the treated females were picked up to mate with WT males to produce embryos (Fig. 4 B). Then the effectiveness of mutations in GCs could be evaluated by dorsal-ventral axis formation defects as reported previously in maternal *hwa* mutants (Yan et al., 2018).

Immunostaining of Cas9 in isolated oocytes from the 4-OHT– or DMSO-treated females after 1 day posttreatment revealed that nCas9$^{ERT2}$ in oocytes could be significantly translocated into the nucleus after 4-OHT treatment (Fig. 4, C and D). At 3 mpf, we collected embryos from seven 4-OHT–treated females and four DMSO-treated females mating with WT males, respectively, and observed that all seven 4-OHT–treated females could produce ventralized embryos, mimicking M*hwa* mutants, with the ratios ranging from 1.71% (2/117) to 14.8% (12/81); meanwhile, no ventralized embryos were produced by four DMSO-treated females (Fig. 4 E). These data imply that *hwa* in some germ stem cells could be mutated to lose function. It is worth noting that the ratio of ventralized embryos was low and varied among individuals. Further optimization of 4-OHT treatments may be needed to achieve a higher gene-editing efficiency. By sequencing *hwa* alleles in individual ventralized and normal embryos produced by 4-OHT– or DMSO-treated transgenic females, we detected different types of *hwa* mutations in embryos produced by 4-OHT–treated females, but no mutations in the *hwa* locus in normal embryos produced by DMSO-treated females (Fig. 4 F and Table S3). Nevertheless, these results support the idea that our inducible nCas9$^{ERT2}$ system works in adult fish.

### Inducible nCas9$^{ERT2}$ system helps determine implication of PGCs-expressed *tbx16* in PGC migration

The zebrafish *tbx16* is initially found to affect movements of mesodermal precursors and somite formation (Ho and Kane, 1990; Kimmel et al., 1989). It is observed later on that *tbx16* mutants also exhibit abnormal locations of PGCs outside the gonad regions (Weidinger et al., 1999), and *tbx16* is also expressed in PGC as early as the dome stage (4.3 hpf) (D'Orazio et al., 2021). This raises a puzzling question: is defective PGC migration in the *tbx16* mutant ascribed to loss of Tbx16 in SCs or in PGCs? We thought that our nCas9$^{ERT2}$ system might help address this question.

Then, we generated the *Tg(U6:tbx16-gRNAs;cryaa:CFP)* transgenic line that expresses two gRNAs for targeting the *tbx16* locus and CFP in the lens for indication of the transgenes (Fig. 5 A). Crosses between *Tg(piwil1:nCas9$^{ERT2}$-n3U;piwil1:mCherry-CAAX-n3U)* females and *Tg(U6:tbx16-gRNAs;cryaa:CFP)* males produced *Tg(piwil1:nCas9$^{ERT2}$-n3U;piwil1:mCherry-CAAX-n3U;U6:tbx16-gRNAs;cryaa:CFP)* triple transgenic embryos, which were treated with 40 μM 4-OHT from 4 hpf onward (Fig. 5 B). As observed at 30 hpf, the treated embryos looked morphologically normal, which was in sharp contrast to the spadetail-like *tbx16$^{tsu-G11s}$* (designated allele code) mutants obtained by ENU-mediated mutagenesis in our lab (Fig. 5 C, left). Genotyping of PGCs within 4-OHT– and DMSO-treated embryos revealed that about 34.6% (9/26) of *tbx16* alleles were mutated in PGCs of 4-OHT–treated embryos, whereas no mutations in the *tbx16* locus were detected in SCs of those embryos (Fig. 5 C, right). Next, we picked up 7–8 single PGCs from each 4-OHT–treated 2 dpf embryo with mCherry expression in PGCs and genotyped each PGC after sequencing of the *tbx16* target sequence (Fig. S4 A). Results revealed that 47.7% (21 of 44) of PGCs from 6 embryos carried *tbx16* mutant alleles, 23.8% (5 of 21) of which were biallelic mutations (Fig. 5 D). Among the mutant alleles, the predominant one was the 31-bp deletion (65.2%) (Fig. S4 B). Furthermore, to validate the knockout efficiency, we checked Tbx16 protein levels by immunostaining in 4-OHT– and DMSO-treated embryos at the bud stage and observed a significant reduction of Tbx16 protein in PGCs from 4-OHT–treated embryos (Fig. 5 E). These data indicate that stage-specific induction of nCas9$^{ERT2}$-based gene-editing activity also works for the *tbx16* locus in PGCs.

To look into the effect of *tbx16* mutations in PGCs, we observed PGC locations in *Tg(piwil1:nCas9$^{ERT2}$-n3U;piwil1:mCherry-CAAX-n3U;U6:tbx16-gRNAs;cryaa:CFP)* embryos at 24 hpf by fluorescent microscopy, which were derived from crosses between *Tg(piwil1:nCas9$^{ERT2}$-n3U;piwil1:mCherry-CAAX-n3U)* females and *Tg(U6:tbx16-gRNAs;cryaa:CFP)* males. The mCherry$^+$-PGCs were normally well aligned in bilateral gonads at the anterior end of the yolk extension at 24 hpf; abnormal location of mCherry$^+$-PGCs occurred in the head or trunk region (Fig. 5 F). The above transgenic embryos of each batch were divided into two groups, one of which was treated with 40 μM 4-OHT from 4 hpf onward and the other one with DMSO. We then counted the number of embryos with obvious ectopic PGCs in the head region at 24 hpf. Results indicated that, on average, the 4-OHT group had a significantly higher proportion of embryos with ectopic PGCs (Fig. 5 G). In addition, we performed *ddx4 in situ* hybridization (ISH) to detect PGC locations in 4-OHT– and DMSO-treated embryos (Fig. S5 C), and the ratio of mislocated PGCs in PGC-specific KO embryos was higher than that of control embryos, supporting the function of Tbx16 in PGC migration. These results suggest that Tbx16 expressed in PGCs plays a role in pledging normal migration of PGCs during somitogenesis, even though the involvement of Tbx16 in SCs in this process cannot be excluded.

### Inducible nCas9$^{ERT2}$ system works in mouse early embryos

To explore the broad applicability of our nCas9$^{ERT2}$ system in other species, we briefly tested blastomere-specific genome editing in mouse early embryos by targeting the *Cdx2* gene, which is specifically expressed in the trophectoderm of blastocysts and represses the expression of *Nanog* and *Oct4* in the inner cell mass (ICM) (Ralston and Rossant, 2008). We co-injected synthetic nCas9$^{ERT2}$ mRNA, *nls-mCherry* mRNA, and two *Cdx2* gRNAs in different combinations into one blastomere of late 2-cell stage

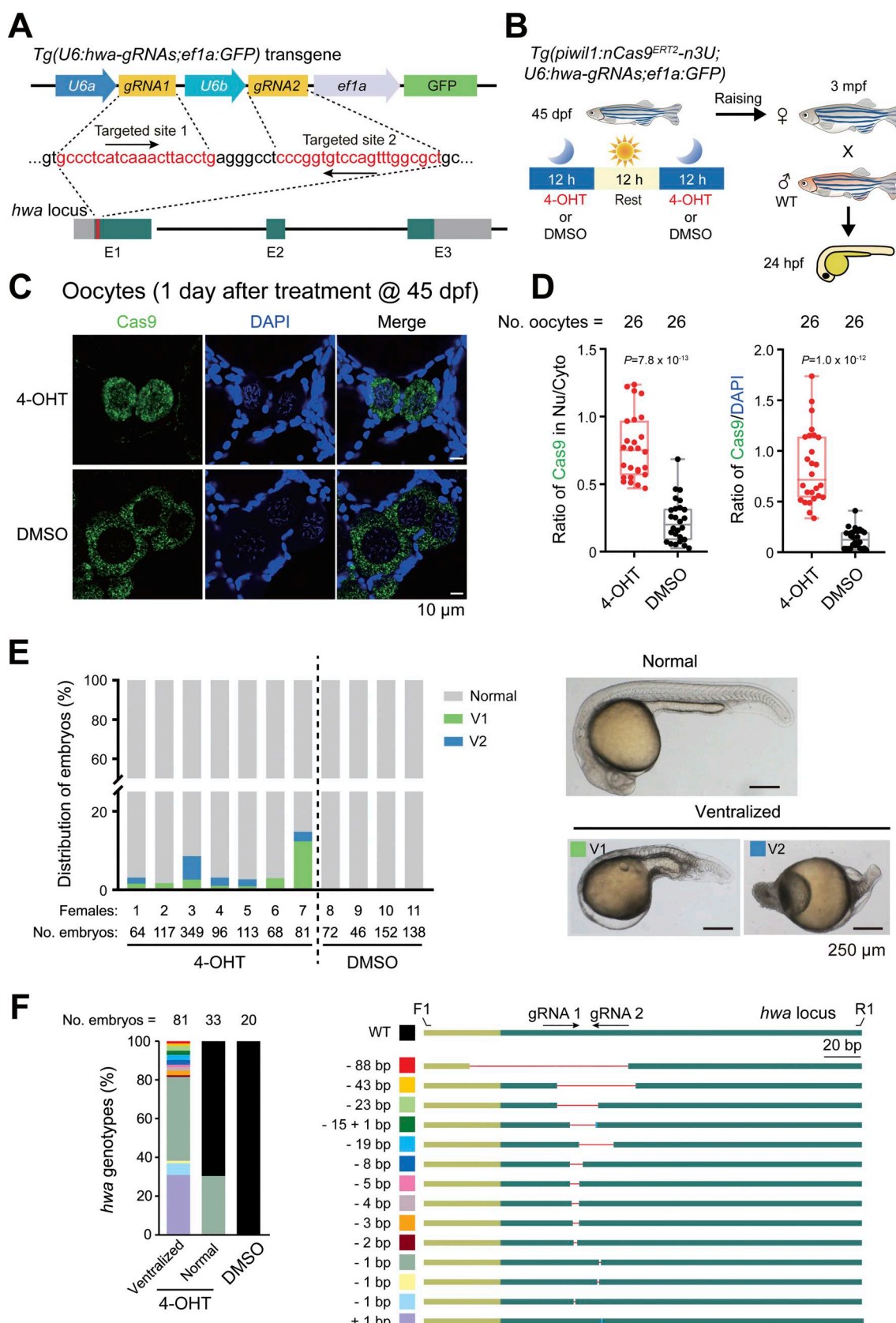

Figure 4. **4-OHT–induced *hwa* mutations in GCs of young females. (A)** Zebrafish *hwa* locus and *hwa-gRNAs* transgene structures. E1–E3, exons; gray bars, UTR. **(B)** Procedure for 4-OHT treatments of the double transgenic young fish and subsequent mating and embryonic analysis. Moon, night; Sun, daytime. **(C)** Immunostaining of Cas9 (green) and nuclei stained by DAPI (blue) in ovaries within the 4-OHT– or DMSO-treated females at 45 dpf. **(D)** Box plots showing

the ratio of Cas9 (green) signal intensity in the nucleus to the cytoplasm of oocytes (left, mean ± SD), and the signal intensity ratio of Cas9 (green) to DAPI (blue) in the nucleus of oocytes (right, mean ± SD). Nu, nucleus. Cyto, cytoplasm. No. oocytes, the number of oocytes. P values of the $t$ test (unpaired, two-sided) were also labeled. **(E)** The ratio of ventralized embryo types (left bar graph) and morphology of representative embryos at 24 hpf (right). Females: the serial number of treated transgenic females. No. embryos, the number of embryos produced by the corresponding females. **(F)** Left, bar chart showing the proportion of different mutated *hwa* genotypes. No. embryos, the number of embryos. Right, schematic showing different mutated genotypes of the *hwa* allele in the ventralized embryos. The light green boxes indicated the UTR of *hwa*, and the dark green boxes indicated the exons of *hwa*. Red lines and blue rectangles represented deletions and insertions, respectively. F1 and R1 are the forward and reverse genotyping primers of the *hwa* target site. Source data are available for this figure: SourceData F4.

embryos, immediately followed by treatment with 10 µM 4-OHT or DMSO (Fig. 6 A). At 24 hpt (at the 8-cell stage), 4-OHT or DMSO was removed by wash, and embryos were either used for isolating mCherry-positive blastomeres for *Cdx2* amplification and sequencing or cultured in fresh medium to the blastocyst stage for cell lineage examination. Sequencing results of pooled mCherry-positive blastomeres of 8-cell stage embryos disclosed that mutant *Cdx2* alleles accounted for 58.82% (10/17) in

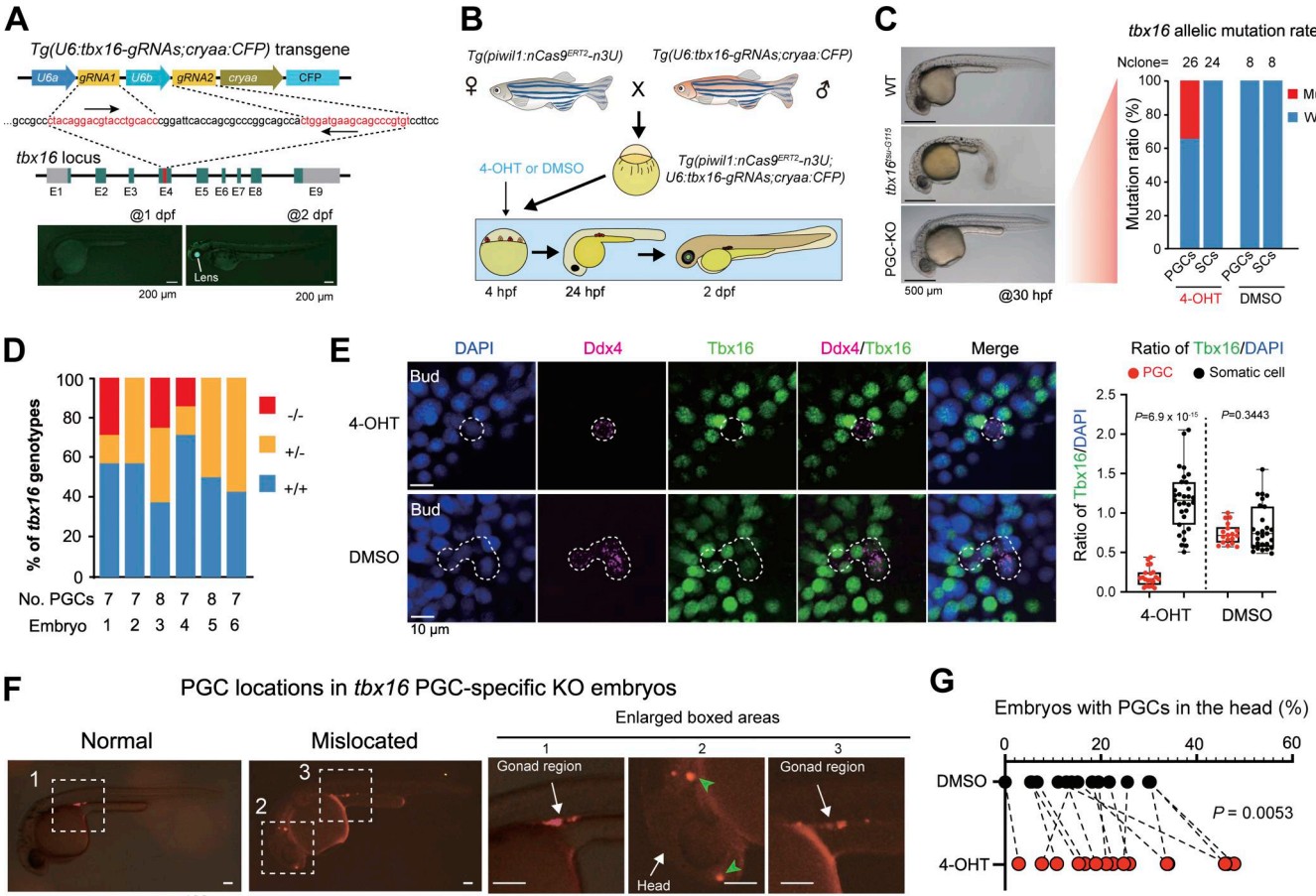

Figure 5. **PGC-specific mutation of *tbx16* affects migration. (A)** Genomic structure of the zebrafish *tbx16* locus and generation of *tbx16-gRNAs* expressing stable germline. The direction of *gRNA* recognition sequences was indicated by arrows. E1–E9, exons; gray areas, UTRs. The low panel showed the transgenic embryos with CFP expression in the lens at 2 dpf but not at 1 dpf. **(B)** Procedures to obtain embryos expressing nCas9$^{ERT2}$ and *tbx16-gRNAs* and to treat the embryos. **(C)** Left, morphology of WT, *tbx16* whole-body knockout mutant (*tbx16$^{tsu-G115}$*) embryos, and *Tg(piwil1;nCas9$^{ERT2}$-n3U;U6:tbx16-gRNAs;cryaa:CFP)* embryos treated with 4-OHT at the sphere stage (PGC-KO). Right, bar chart showing genotypes of PGCs and SCs in DMSO and 4-OHT–treated groups at 24 hpf. Nclone, the number of monoclones for Sanger sequencing. **(D)** Ratio of PGC genotypes in PGC-KO embryos as in B. Individual PGCs were isolated from individual 4-OHT–treated *Tg(piwil1;nCas9$^{ERT2}$-n3U;piwil1:mCherry-CAAX-n3U;U6:tbx16-gRNAs;cryaa:CFP)* embryos at 2 dpf and subjected to amplification, cloning, and sequencing of the *tbx16* target region, allowing identification of each PGC's genotype. –/–, two mutant alleles; +/–, one mutant allele; +/+, two WT alleles. No. PGCs, the number of PGCs within the corresponding embryos. Embryo: the serial number of 4-OHT–treated transgenic embryos. **(E)** Left, immunostaining results of Tbx16 (green) and Ddx4 (magenta) with nuclei stained by DAPI (blue) in 4-OHT– or DMSO-treated embryos from the sphere stage. Scale bar, 10 µm. Right, box plot showing the statistical results of the ratio of Tbx16 signals to DAPI in PGCs and SCs in either 4-OHT– or DMSO-treated embryos (mean ± SD). Each independent data point represents a cell. The numbers of PGCs and SCs in the 4-OHT–treated group are 20 and 31, respectively, while the numbers of PGCs and SCs in the control group are 18 and 27. P values of the $t$ test (unpaired, two-sided) were also labeled. **(F)** Examples of PGC locations in 24 hpf embryos. The boxed areas in the left panel were enlarged in the right panels. In the second boxed area, PGCs were apparently mislocated in the head region. **(G)** Ratio of embryos with PGCs in the head in 4-OHT–treated embryos (as shown in F) and DMSO-treated control embryos. Each independent data point represents an independent experiment, totaling 15 times. P values of the $t$ test (paired, two-sided) were also labeled. Source data are available for this figure: SourceData F5.

blastomeres derived from embryos injected with all three RNA species and treated by 4-OHT (M group). In contrast, no mutant alleles were detected in blastomeres derived from embryos injected similarly but treated with DMSO (C3 group) (Fig. 6 B). Genotyping individual mCherry-positive blastomeres revealed a predominance of biallelic mutations and different types of mutations (Fig. 6, C and D) in M group embryos. Immunostaining of embryos at the blastocyst stage showed that the M group had a higher proportion of mCherry-positive blastomeres with loss of Cdx2 expression than the other control groups, although the ratios of mCherry-positive cells per embryo were comparable among the M group and control groups (C1, C2, and C3) (Fig. 6, E and F). Consistent with the idea that Cdx2 is a suppressor of *Nanog* transcription (Ralston and Rossant, 2008), strikingly, the percentage of Nanog⁺;mCherry⁺-blastomeres increased in the M group embryos (Fig. 6, G and H). These observations together suggest that the inducible nCas9$^{ERT2}$ system is applicable to mouse embryos for gene knockout.

## Discussion

In this study, we established an inducible gene-editing approach to perform spatiotemporal gene knockout based on 4-OHT–inducible nCas9$^{ERT2}$. As a proof of concept, we applied this system to GC-specific gene knockout by expressing nCas9$^{ERT2}$ in GCs driven by the GC-specific *piwil1* promoter, expressing gene-specific gRNAs ubiquitously, and stage-specific 4-OHT treatment.

Our inducible gene-editing system is composed of two major components: a transgenic line with tissue-specific expression of 4-OHT–inducible nCas9$^{ERT2}$ as an acceptor and a transgenic line with ubiquitous expression of gene-specific gRNAs. The latter could be replaced by the injection of synthetic gRNAs. Once a tissue-specific transgenic line with strong nCas9$^{ERT2}$ expression is established, it could be used for performing gene knockout of many genes, even for simultaneous knockout of several genes to study their functional redundancy. This is particularly useful in the zebrafish, in which loxP knock-in usually has a very low efficiency. In case several genes need to be knocked out simultaneously to study their functional redundancy, the application of the Cre/loxP system has to first generate at least one loxP knock-in line for each gene and then intercross them several times to obtain animals with multiple loxP knock-ins, which is usually time-consuming and often impractical. In contrast, the nCas9$^{ERT2}$ system only needs one nCas9$^{ERT2}$ transgenic line plus co-injection of several genes' gRNAs for simultaneous knockout of these genes.

In our inducible system, nCas9$^{ERT2}$ nuclear translocation relies on the action of 4-OHT. The concentration and treatment duration of 4-OHT should be optimized on a case-by-case basis. Actually, we employed two types of 4-OHT dosing selection schemes for our system, tailored to the developmental stages. For early embryogenesis, we tested four concentrations of 4-OHT: 25, 30, 40, and 50 μM from the sphere to the 2 dpf stages and detected the survival rates of treated embryos at the 2 dpf, indicating a severe cytotoxic effect with the 50 μM treatment (Fig. S5, A and B). In addition, by the T7EI assay, we found that

the genome-editing efficiency of 40 μM 4-OHT treatment was higher than that of 25 μM (Fig. S5 C). Via immunostaining of Cas9 protein, we revealed that 40 μM 4-OHT treatment was sufficient to trigger nCas9$^{ERT2}$ nuclear translocation in PGCs (Fig. S5 D). Thus, a concentration of 40 μM was used for early embryos in the subsequent experiments. Next, for adult treatment, we performed a spawning rate detection by 5 or 10 μM 4-OHT treatment (Fig. S5 E) and observed a severe disruption of reproduction in the 10 μM 4-OHT group (Fig. S5 F). Thus, although the relatively lower editing efficiency, we still chose a concentration of 5 μM for adult treatment to ensure the spawning rate. We noted that nCas9$^{ERT2}$-mediated gene mutation does not occur in all cells of a target tissue, which may be ascribed to differential accessibility to the target gene locus in different cell types or cell statuses. Besides, only some target cells carry mutations in both alleles, which may make phenotypic analysis complicated. Nevertheless, the inducible nCas9$^{ERT2}$ system would be very useful to explore stage-specific and/or tissue-specific function of a gene whose mutation in a whole organism may lead to death or pleiotropic effects.

Around the shield stage, four clusters of PGCs begin migrating toward the dorsal side of embryos, requiring chemokine signal guidance from SCs (Aguero et al., 2017; Raz, 2003). PGCs rely on the intracellular chemokine receptor Cxcr4b to respond to the chemokine Cxcl12a (also known as Sdf1) expressed by surrounding SCs, migrating directionally toward regions with a higher level of Cxcl12a (Boldajipour et al., 2008; Doitsidou et al., 2002; Paksa and Raz, 2015; Weidinger et al., 1999). Knockdown or knockout of Cxcr4b or Cxcl12a results in an obvious incorrect migration of PGCs (Doitsidou et al., 2002; Molyneaux et al., 2003). Tbx16 has also been reported to participate in regulating the migration of PGCs. In 24 hpf *tbx16* zygotic mutant (Z*tbx16*) embryos, a portion of PGCs is ectopically localized in the head or tail region of embryos (Weidinger et al., 1999). The authors speculated that this phenotype is caused by a deficiency of Tbx16 protein, which causes an aberrant expression pattern of *cxcl12a* in SCs and disrupts PGC migration. Cxcl12a/b and Cxcr4a are also involved in SC migration during gastrulation (Mizoguchi et al., 2008). During zebrafish endodermal development, the Tbx16 protein might directly act on transcription of *cxcl12a/b*, which is downregulated in the Z*tbx16* embryos at the 75% epiboly stage (Garnett et al., 2009; Nelson et al., 2017). Another study also reported that the expression pattern of *cxcl12a* is altered in Z*tbx16* embryos (Weidinger et al., 1999).

Considering that expression levels of *tbx16* in PGCs and SCs are basically the same throughout early embryogenesis (Skvortsova et al., 2019), whether *tbx16* expressed in PGCs participates in PGC development and migration remains to be explored. This work offers an alternate explanation for the abnormal PGC migration phenotype observed in Z*tbx16* embryos. With this inducible gene-editing system, we tissue specifically knocked out *tbx16* in PGCs to avoid potential interference from *tbx16* deficiency in SCs on PGC migration. We found that PGC-specific knockout of *tbx16* disrupted PGC migration, resulting in a significant increase in the proportion of embryos with ectopic PGCs. This result indicates that *tbx16* expressed in PGCs plays a cell-autonomous role in the PGC

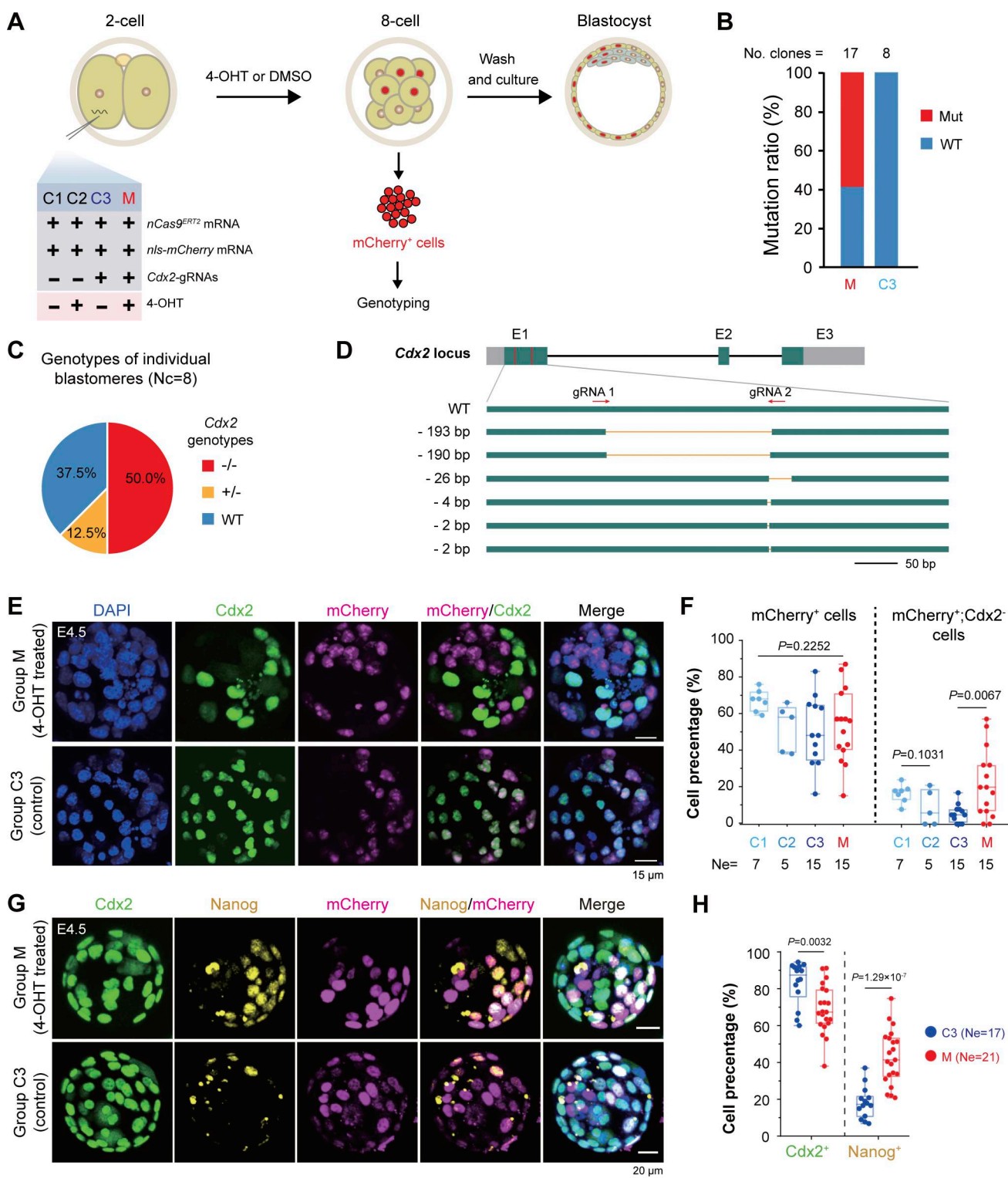

Figure 6. **Mutation of *Cdx2* in selected blastomeres in mouse early embryos using inducible nCas9ERT2. (A)** Experimental procedure. Different combinations (groups C1–C3 and M) of indicated reagents were injected into one blastomere of 2-cell stage embryos. **(B)** Mutation ratios of *Cdx2* alleles in mCherry+ cells of Group M or Group C3 embryos at the 8-cell stage. A *Cdx2* region, including gRNA target sites, was amplified from pooled cells and cloned for sequencing. The number of sequenced clones was indicated. Mut, mutant alleles; WT, WT allele. **(C)** Ratios of cells with specific *Cdx2* genotypes in group M embryos. The mCherry+ cells were isolated and genotyped individually. –/–, two mutant alleles; +/–, one mutant allele; WT, two WT alleles. Nc, the number of genotyped cells. **(D)** Structure of WT and mutant alleles as revealed in C. E1–E3, exons; gray, UTRs; arrows, direction of *gRNA* recognition sites; orange line, deleted region. **(E and F)** Representative immunofluorescent images of Cdx2 and mCherry expression (E) and ratio of cells with or without indicated markers (F) in different groups of blastocysts at E4.5, shown as (mean ± SD). **(G and H)** Representative immunofluorescent images of Cdx2, Nanog, and mCherry expression (G) and ratio of cells with Cdx2 or Nanog expression (H) in group M or C3 blastocysts at E4.5, shown as (mean ± SD). In F and H, the denominator was the total number of cells in individual blastocysts; P values of one-way ANOVA in the left of (F) and t test (unpaired, two-sided) in other boxplots in F and H were indicated; Ne, the number of blastocysts. Source data are available for this figure: SourceData F6.

migration process, and its absence increases the likelihood of incorrect PGC migration.

## Materials and methods

### Breeding of zebrafish

The WT Tübingen strain and all transgenic zebrafish lines used in this study were normally raised at 28.5°C in standard housing systems, with ethical approval from the Animal Care and Use Committee of Tsinghua University. Embryo developmental stages were defined as standard developmental stages (Kimmel et al., 1995). The generation of transgenic lines is described below.

### Construction of plasmids

In this work, we constructed three types of plasmids for mammalian cell transfection, transgenic zebrafish line construction, and in vitro transcription of mRNA, respectively. All of the transgenic plasmids in this research were constructed with *Tol2* transposon–based vectors. Table S1 lists the plasmid information about vectors and restriction enzyme sites for insertions. Table S2 presents the primers used for the inserted sequences in these plasmids, including four kinds of *piwil1* promoters, *cryaa* promoters, and *ef1a* promoters. The majority of the plasmids were constructed by the Gibson Assembly method (Avilan, 2023), except for *pU6a-EMX1 gRNA-Cas9-T2A-GFP*, which was ligated with T4 ligase following restriction enzyme cleavage. In brief, the Gibson Assembly method involves adding the same amount of DNA sequences with overlapping homologous arms to an EP tube, followed by 2X MultiF Seamless Assembly Mix (RK21020; ABclonal), then incubating at 50°C for about 1 h. The *U6a:gRNA* fragment in the *pU6a-EMX1 gRNA* plasmid originated from *pU6a-EMX1 gRNA-Cas9-T2A-GFP*.

### Cell culture and transfection

HEK293T cells (purchased from Cell Resource Center, Peking Union Medical College, Beijing, China) were cultured in DMEM supplemented with 10% fetal bovine serum (FBS) and penicillin/streptomycin at 37°C in a humidified incubator with 5% $CO_2$. When the cell density is about 50%, plasmid transfection could be performed using VigoFect (T001; Vigorous). Plasmid transfection requires 2 µg per well in a 6-well cell culture plate. The mixture of transfection reagent and plasmids was obtained according to the manufacturer's instructions and then added into the well. After 8 h, replace with fresh DMEM.

### Generation of transgenic zebrafish lines

After plasmid construction, 30 pg of plasmid and 200 pg of *Tol2* transposase mRNA were co-injected into WT zygotes. For transgenic lines with fluorescence, embryos were screened out by observing fluorescence with a fluorescent stereomicroscope (MVX10; Olympus) at 1 or 2 dpf. *Tg(piwil1:nCas9^{ERT2}-n3U)* and *Tg(piwil1:nCas9-n3U)* transgenic embryos did not express fluorescence, so these embryos were screened by genotyping and ISH to detect *Cas9* mRNA expression.

### Mouse embryo culture

The mouse strain C57BL/6N was used in this study. Mice were maintained on a 12/12-h light/dark cycle at 22–26°C, with sterile pellet food and water *ad libitum*. 12–14 g female mice were superovulated by intraperitoneal injection of 10 IU pregnant mare serum gonadotropin (21958956; PMSG, Sansheng Biological Technology) and 10 IU human chorionic gonadotrophin (110041282; HCG; Sansheng Biological Technology) at the interval of 47.5 h. Then, the female mice were mated with 8- to 20-wk-old male mice one to one. Embryos were collected from the oviduct of female mice with a vaginal plug in the morning, and granule cells were removed by digestion with 300 µg/ml hyaluronidase (H4272; Sigma-Aldrich) in M2 media (MR-015-D; Merck Millipore). Fertilized embryos with the second polar body were picked out and cultured in a 30 µl KSOM (MR-121-D; Merck Millipore) droplet covered by mineral oil (M8410; Sigma-Aldrich) in a 37°C incubator filled with 5% $CO_2$. Developmental stages of mouse embryos were defined as follows: early 1-cell stage, 18–21 hpi (hours post HCG injection) or 9–12 hpf; late 2-cell stage, 48–50 hpi or 39–41 hpf.

### 4-OHT treatment of zebrafish and mouse embryos

The 4-OHT (H7904; Sigma-Aldrich) is dissolved in DMSO to 10 mM and stored at –20°C, kept out of the light. When treating zebrafish, 4-OHT is diluted with Holtfreter water (for embryos and larvae) or water from the housing system (for adults) to the proper concentration and kept away from light. When confirming whether the nCas9^{ERT2} protein could respond to 4-OHT treatment, embryos at the sphere stage were incubated in 6-well plates with 3 ml of 20 µM 4-OHT. When detecting the mutation efficiency of target genes, the embryos were incubated in 6-well plates with 3 ml of 40 µM 4-OHT to induce a higher mutation efficiency. The treatment began at the sphere stage and ended at 48 hpf. The young zebrafish at 45 dpf were incubated in 30 ml of 5 µM 4-OHT. When treating mouse embryos, 4-OHT is diluted with KSOM media to 10 µM and kept out of the light.

### Immunofluorescence staining of HEK293T cells, zebrafish embryos and oocytes, and mouse embryos

HEK293T cells, zebrafish embryos, and mouse embryos were all fixed with 4% paraformaldehyde (P6148; Sigma-Aldrich) in PBS at 4°C. The HEK293T cells were fixed for 30 min, whereas the zebrafish and mouse embryos were fixed overnight. Zebrafish embryos after 24 hpf were dechorionated with pronase before fixation, while embryos before 24 hpf were mechanically dechorionated after fixation. Zebrafish embryos were then dehydrated in 100% methanol with a graded PBS:methanol series (3:1, 1:1, and 1:3) and stored at –20°C, ready for immunofluorescence experiments. Stained HEK293T cells and embryos were imaged with an Olympus FV3000 microscope. The confocal data were processed using Imaris software. Immunofluorescence staining was carried out as the standard protocol (Köktürk and Altındağ, 2024; Sorrells et al., 2013). Briefly, HEK293T cells were permeabilized in 0.4% Triton X-100 in PBS for 10 min, washed with PBS three times for 10 min each, blocked with 3% BSA in PBS for 30 min, and treated with mouse anti-Cas9 (1:200 dilution, Ab191468; Abcam) as the primary antibody overnight at 4°C. The next day, HEK293T cells were washed with PBS three times for 10 min each, incubated in DAPI (1:10,000 dilution; Thermo Fisher Scientific) and fluorochrome-conjugated secondary antibodies in

blocking buffer (1:200 dilution) for 2 h, then washed with PBS three times for 10 min each and prepared for imaging. All subsequent steps after DAPI and secondary antibodies were kept in the dark. The secondary antibodies included Alexa Fluor 488 AffiniPure Goat Anti-Mouse IgG (H + L) (115-545-003; Jackson ImmunoResearch Labs) and Rhodamine (TRITC) AffiniPure Goat Anti-Rabbit IgG (H + L) (111-025-003; Jackson ImmunoResearch Labs).

After rehydration in a graded methanol:PBS series (3:1, 1:1, 1:3, and 0:4), zebrafish embryos before 24 hpf were permeabilized in 0.5% PBST for 30 min, whereas embryos at 24 hpf or later were permeabilized in 1% PBST overnight. After that, embryos were treated for antigen retrieval with preheated 10 mM sodium citrate (pH 6.0), maintained at 95°C for 10 min, and cooled on a benchtop for at least 30 min. The embryos were then washed with PBS three times for 10 min each, blocked with 10% BSA, and treated with primary and secondary antibodies diluted in blocking buffer, similar to the experiment methods for HEK293T cells, except that the incubation of DAPI and fluorochrome-conjugated secondary antibodies was overnight at 4°C, and they were washed with PBS six times for 10 min each after primary and secondary antibody incubation. The primary antibodies included mouse anti-Cas9 (Ab191468, 1:100 dilution; Abcam), rabbit anti-Ddx4 (GTX128306-S, 1:1,000 dilution; GeneTex), rabbit anti-Tbx16 (GTX128406-S, 1:1,000; GeneTex), mouse anti-Ddx4 (1:500), and rabbit anti-mCherry (BE2027, 1:200 dilution; Easybio). The secondary antibodies included Alexa Fluor 647 AffiniPure Goat Anti-Rabbit IgG (H + L) (111-605-003; Jackson ImmunoResearch Labs), Rhodamine (TRITC) AffiniPure Goat Anti-Rabbit IgG (H + L) (111-025-003; Jackson ImmunoResearch Labs), and Alexa Fluor 488 AffiniPure Goat Anti-Mouse IgG (H + L) (115-545-003; Jackson ImmunoResearch Labs). Immunostaining of Cas9 protein was performed using the tyramide signal amplification (TSA)-based immunostaining method (Faget and Hnasko, 2015). Briefly, zebrafish embryos were treated with 1% $H_2O_2$ (1% $H_2O_2$ in PBS) for 30 min after rehydration and kept out of the light, then underwent permeabilization, antigen retrieval, blocking, and primary antibody incubation, similar to the experiment methods for other antibodies. The embryos were then incubated with DAPI and an HRP-conjugated goat anti-mouse antibody (1:200 dilution; BE0102-100; Easybio) overnight at 4°C, which served as the secondary antibody for Cas9. The next day, embryos were washed with 0.1% Tween-80 in PBS (0.1% PBST) three times for 10 min each and three times for 20 min each, then incubated in the TSA reagent (1:100; TSA-Cy5; PerkinElmer) as the reaction substrate for HRP-conjugated antibody at room temperature for 1 h and kept away from light. Finally, the embryos were washed three times with 0.1% PBST, each for 20 min.

After anesthetization and euthanasia of the female fish, ovaries were surgically excised, mechanically dissociated into small clusters by forceps, and fixed in 4% paraformaldehyde overnight at 4°C. The samples were subsequently dehydrated in methanol at –20°C and rehydrated in 0.2% PBST. They were then incubated in 1% PBST at room temperature for 30 min, followed by antigen retrieval in 10 mM sodium citrate buffer (pH 6.0) at 95°C for 10 min. The antibodies were used as follows: The primary antibody was mouse anti-Cas9 (1:100 dilution; Ab191468;

Abcam). The secondary antibodies were Alexa Fluor 488 AffiniPure Goat Anti-Rabbit IgG (H + L) (1:200 dilution, 111-545-003; Jackson ImmunoResearch Labs) with DAPI (C0060; Solarbio).

Mouse embryos were permeabilized in 0.5% PBST for 30 min, and the subsequent experiment procedures were similar to those of zebrafish embryos but without antigen retrieval. The primary antibodies included mouse anti-Cdx2 (1:200 dilution, MU392A-UC; BioGenex), rabbit anti-Nanog (1:100 dilution, Ab214549; Abcam), and rabbit anti-mCherry (1:200 dilution, BE2027; Easybio). Using the same secondary antibodies as mentioned above.

### Design and test of gRNAs
The gRNAs of target genes in this study were designed on the CHOPCHOP website (https://chopchop.cbu.uib.no/), and the genotyping primers were also designed on this website. To test the gRNA mutation efficiency, 300 pg of gRNA and 200 pg of *Cas9* mRNA were co-injected into WT zygotes. *Cas9* mRNA was zebrafish codon optimized and transcribed *in vitro* with the mMESSAGE mMACHINE T7 Transcription kit (AM1344; Thermo Fisher Scientific), and gRNAs were transcribed in vitro with the MEGAscript T7 Kit (AM1333; Thermo Fisher Scientific). The mutation efficiency of gRNA was determined by genotyping and T7E1 assay with injected embryos at 24 hpf. Table S2 lists the primers used for gRNA generation.

### Collection of zebrafish PGCs and SCs and mouse blastomere
For 24 hpf and 2 dpf zebrafish embryos, PGC-containing regions of embryos were excised and transferred into 1.5-ml Eppendorf tubes. 10–50 µl of trypsin was added per tube, and tubes were placed at 37°C for digestion. The mixture was pipetted 20 times through 200 µl tips every 10 min until no visible tissue remained. Cell suspensions were then transferred to clear and RNase-free Petri dishes and mCherry-labeled PGCs were manually picked out under a fluorescence microscope with a mouth pipette. mCherry-negative cells are SCs utilized to determine if the target genes are mutated in SCs.

For 8-cell stage mouse embryos, the zona pellucida of the embryos was eliminated using acid Tyrode's solution (MR-004-D; Merck Millipore). Then, the embryos were immediately transferred to 0.25% Trypsin (0458; Amresco) at 37°C, separated into single blastomeres by gentle pipetting, and washed three times in KSOM media, which took ~5–10 min. Blastomeres were transferred to clear KSOM media, and mCherry-positive blastomeres were manually picked out with a mouth pipette under an Olympus inverted fluorescence microscope.

### Genome extraction and genotyping
For genome extraction of HEK293T cells, about 20,000 cells were collected and lysed in 15 µl of 50 mM NaOH at 95°C for 10 min, followed by the addition of 1/10 volume of 1 M Tris-HCl (pH 8.0) and vortexing. The same procedure was employed to extract the genomes of zebrafish and mouse embryos. A 24 hpf zebrafish embryo was lysed with 30 µl of 50 mM NaOH at 95°C for 20 min, while the mouse blastocyst was lysed with 10 µl of 50 mM NaOH at 95°C for 10 min. The genome of adult zebrafish was extracted using the same method with clipped caudal fin tissue. The lysate would then be used as a genome template for

PCR amplification. The genomes of PGC mixtures, individual PGCs, and individual mouse blastomeres were extracted and amplified with the Single Cell WGA Kit (N603; Vazyme), diluted with ddH$_2$O to a suitable concentration, and used as PCR templates. Table S2 summarizes the primer sequences for genotyping.

When detecting the target gene mutations in HEK293T cells, embryos, and cell mixtures, PCR products were further examined by T7EI (NEB) digestion at 37°C for 30 min, followed by sequencing for determination of the mutated sequence. The mutant gene would result in double bands after T7EI digestion and dual peaks in the Sanger sequencing map. To calculate the allelic mutation ratios, the PCR products of gRNA-targeted regions were ligated to the pBM18A vectors (CL073; Biomed) for 2 h at room temperature, then transformed into competent *Escherichia coli*. The next day, monoclones were picked out for Sanger sequencing. When genotyping a single PGC or mouse blastomere, the PCR products were also ligated to the pBM18A vectors for monoclonal Sanger sequencing to determine whether the PGC was hetero-mutated or homo-mutated.

### NGS of zebrafish PGCs and SCs

For the detection of mutation efficiency within this system, PGCs and SCs of zebrafish embryos were collected via fluorescence-activated cell sorting (FACS) and subjected to whole-genome amplification (WGA) using the Single Cell WGA Kit (N603; Vazyme). The target region of *tbxta* was subsequently amplified from the WGA products by PCR using primers ligated with index and adapter sequences required for NGS. The primer sequences are presented in Table S2. Sequencing data were analyzed with CRISPResso2 (Clement et al., 2019), with base substitutions being ignored to minimize potential noise introduced during PCR amplification.

For validation of off-targeting effects within this system, PGCs of zebrafish embryos were also collected via FACS and subjected to WGA using the Single Cell WGA Kit (N603; Vazyme), followed by genomic DNA library preparation using the DNA Library Prep Kit for Illumina (TD503; Vazyme). The DNA libraries were then sequenced with WGS. Potential off-target sites for each gRNA were predicted using Cas-OFFinder (Bae et al., 2014), allowing up to 4 mismatches. Genotypes at each target site were manually curated.

### ISH

The steps of zebrafish embryo fixation and dehydration are consistent with those in immunofluorescence staining. ISH was carried out according to the standard protocols as reported (Chitramuthu and Bennett, 2013; He et al., 2020). In brief, after rehydration in a graded methanol:PBS series (3:1, 1:1, 1:3, and 0:4), zebrafish embryos were fixed with 4% PF for 20 min at room temperature, washed with nuclease-free (NF) 0.1% PBST three times for 5 min each, pre-hybridized in NF HYB- solution (formamide:20× SSC:NF ddH$_2$O = 2:1:1) at 65°C for at least 4 h, and then hybridized with DIG-labeled probe in NF HYB+ solution (HYB- solution with 1,000× heparin and 100× yeast RNA) at 65°C overnight. The next day, embryos were washed at 65°C with 2× SSCT (0.1% Tween-80 in 2× SSC)/50% formamide twice for 30 min, 2× SSCT for 15 min, and 0.2× SSCT twice for 30 min;

blocked with blocking solution (goat serum: block reagent: MAB = 1:2:7) for at least 1 h; and incubated with Anti-Digoxigenin AP antibody (1:3,000 dilution, 11093274910; Roche) at 4°C overnight. On the third day, embryos were washed with 0.1% Tween-80 in MAB six times for 25 min and finally incubated in BM Purple AP substrate (11442074001; Roche) with 5 mM levamisole added for detecting hybridization signals. The anti-*ddx4* probe was synthesized using Roche DIG RNA Labeling Mix (11277073910). ISH-stained embryos were imaged using a Nikon stereomicroscope.

### Microscopy imaging

The immunofluorescence staining images of embryos were acquired by an Olympus FV3000 confocal microscope. Embryos were placed in glass-bottomed culture dishes and scanned at room temperature (about 24°C). To stabilize the positions of embryos, zebrafish embryos were embedded in 1% low-melting agarose (0815; Amresco) gel, whereas mouse embryos were placed in KSOM medium. Immunofluorescence staining images were acquired with FV31S (Olympus) software and processed with Imaris (Oxford Instruments) software. The immunofluorescence staining images of zebrafish ovaries were acquired by a Zeiss LSM980 Airyscan2 confocal microscope. Ovaries were also placed in glass-bottomed culture dishes, embedded in 1% low-melting agarose (0815; Amresco) gel, and scanned at room temperature (about 24°C) with Zen (Zeiss) software.

The fluorescence and white light images of zebrafish embryos were captured by an Olympus MVX10 fluorescent stereomicroscope with an MVPLAPO 1× objective lens (magnification 0.63×–6.3×, numerical aperture 0.25) and an Olympus DP80 camera. Zebrafish embryos were placed in 5% methylcellulose (Sigma-Aldrich) to maintain their morphology, and image acquisition was carried out at room temperature (about 25–30°C). The image acquisition software is cellSens Standard (Olympus).

### Statistical analysis

An average from multiple samples was expressed as mean ± SD. The significance of the difference between groups was analyzed by Student's *t* test (unpaired, two-tailed), with the exception of Student's *t* test (paired, two-tailed) in Fig. 5 G and one-way ANOVA in Fig. 6 F (left). Significant levels were indicated in the corresponding context. Data distribution was assumed to be normal, but this was not formally tested.

### Online supplemental material

Fig. S1 shows the nuclear translocation ability of Cas9 protein expressed in HEK293T cells. Fig. S2 shows the GC specificity of zebrafish *piwil1* promoter. Fig. S3 shows off-target analysis of the *tbxta* locus with WGA and NGS. Fig. S4 shows mutation efficiency of *tbx16* alleles in PGCs. Fig. S5 shows 4-OHT dosing selection for embryos and adults. Table S1 shows plasmid information. Table S2 shows primer information. Table S3 shows monoclonal Sanger sequencing results of *hwa* alleles in embryos.

### Data availability

The raw WGA sequence data reported in this paper have been deposited in the Genome Sequence Archive in the National Genomics Data Center, China National Center for

Bioinformation/Beijing Institute of Genomics, Chinese Academy of Sciences (GSA: CRA033005), which are publicly accessible at https://ngdc.cncb.ac.cn/gsa.

## Acknowledgments

We thank all members of the Meng lab for their intellectual and technical support. We would like to express our gratitude to Dr. Ming Shao for providing *pU6:gRNA* plasmids and Dr. Dahua Chen for providing the mouse anti-Ddx4 antibody. We are grateful to the Cell Biology Facility and the Sharing Core Facility affiliated with the Center of Biomedical Analysis, Tsinghua University, for technical assistance and daily equipment support.

This work is financially supported by the Excellent Research Group Program of NSFC (#3258820001 to A. Meng), the National Key Research and Development Program of China (#2023YFA1800300 to X. Wu and 2018YFC1003304 to A. Meng), and the Yunnan Provincial Science and Technology Project at Southwest United Graduate School (#202302AO370011 to A. Meng).

Author contributions: Yaqi Li: formal analysis, investigation, methodology, validation, visualization, and writing—original draft, review, and editing. Weiying Zhang: data curation, investigation, methodology, resources, and validation. Zihang Wei: data curation, formal analysis, investigation, methodology, validation, visualization, and writing—review and editing. Han Li: data curation, formal analysis, investigation, methodology, validation, visualization, and writing—original draft. Xin Liu: data curation, formal analysis, investigation, and visualization. Tao Zheng: resources. Tursunjan Aziz: formal analysis. Cencan Xing: data curation, methodology, and supervision. Anming Meng: conceptualization, data curation, funding acquisition, project administration, supervision, visualization, and writing—review and editing. Xiaotong Wu: conceptualization, funding acquisition, project administration, supervision, and writing—original draft, review, and editing.

Disclosures: The authors declare no competing interests exist.

Submitted: 28 December 2024

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

# Supplemental material

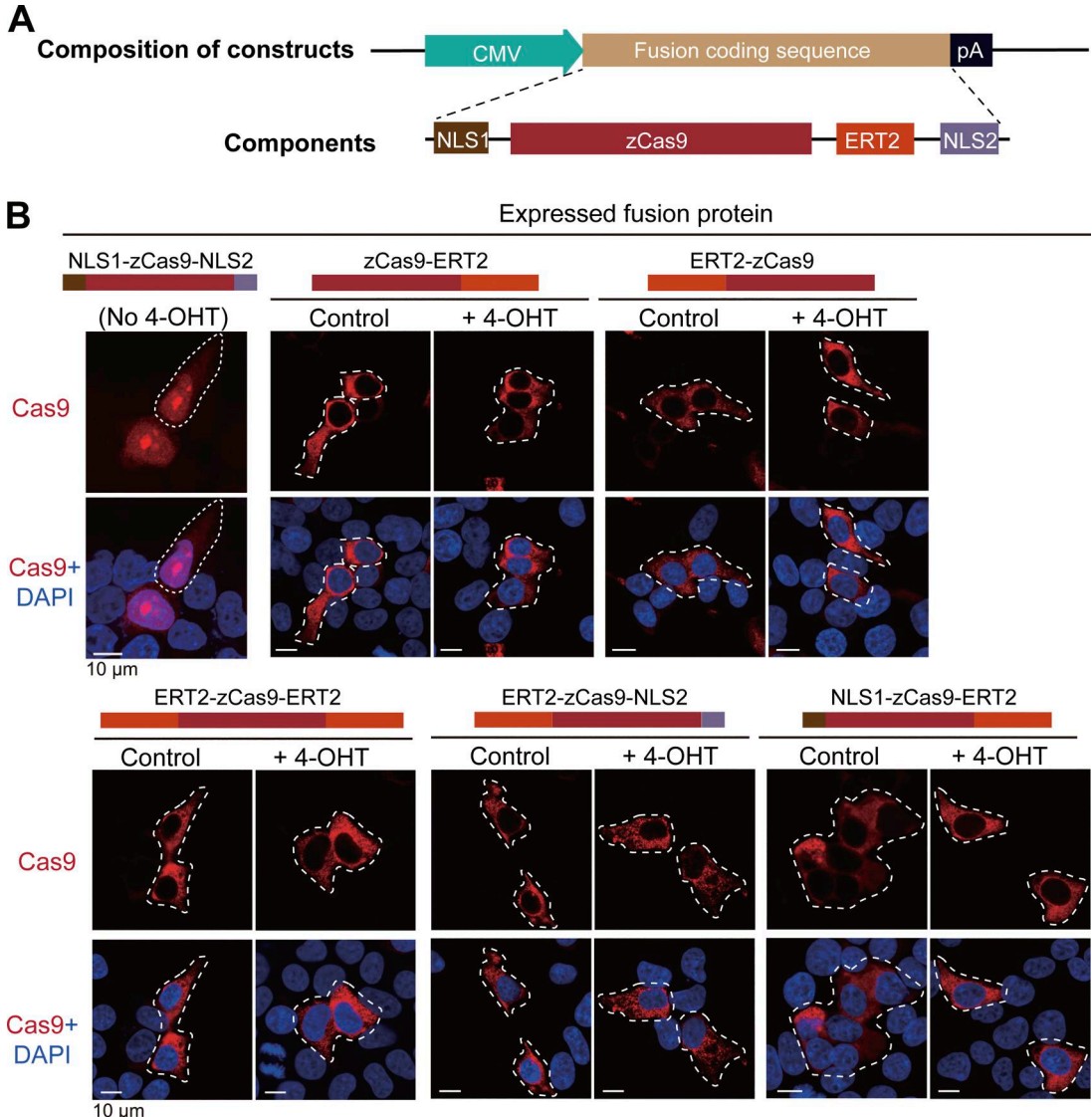

Figure S1. **The nuclear translocation ability of Cas9 protein expressed in HEK293T cells. (A)** General structure of transfection constructs. CMV, cytomegaloviral promoter; zCas9, Cas9 with codon optimization for the zebrafish; ERT2, human estrogen receptor with three missense mutations (G400V/M543A/L544A) in the ligand-binding domain; NLS1, SV40 nuclear localization signal; NLS2, nucleoplasmin NLS; pA, polyadenylation signal. **(B)** Nuclear translocation ability of different Cas9 fusion proteins with or without 4-OHT treatment. The transfected cells were immunostained with Cas9 antibody plus DAPI staining for nuclei 24 hpt. The top of each group of images showed the schematic structure of the Cas9 fusion protein. Source data are available for this figure: SourceData FS1.

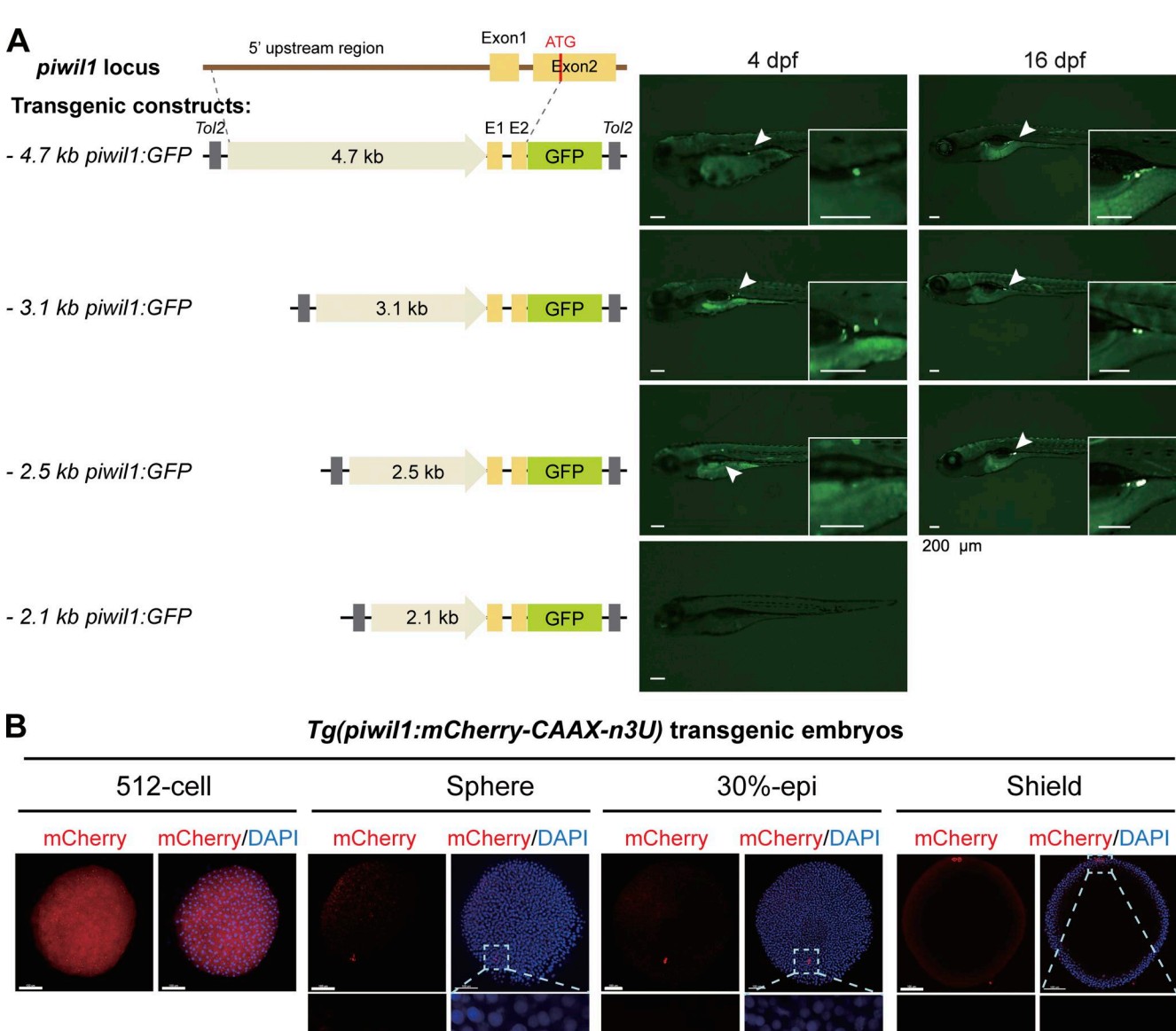

Figure S2. **GC specificity of the zebrafish *piwil1* promoter. (A)** Identification of a shorter *piwil1* promoter with GC specificity. Left, *Tol2* transposon–based construct structures; right, representative GFP expression patterns in zebrafish larvae after injection with corresponding constructs. GFP was observed under fluorescent dissection microscopy. Note that the 2.5-kb *piwil1* promoter retained GC-specific transcription activity and was used subsequently. **(B)** A transgenic line with GC-specific expression of membrane-localizing mCherry driven by the 2.5-kb *piwil1* promoter. CAAX, cell membrane localizing signal; n3U, 3' UTR of the zebrafish *nanos3* gene. Note that mCherry was ubiquitously expressed at the 512-cell stage but became restricted to PGCs thereafter. The boxed areas in the top panel were enlarged in the lower panel. Source data are available for this figure: SourceData FS2.

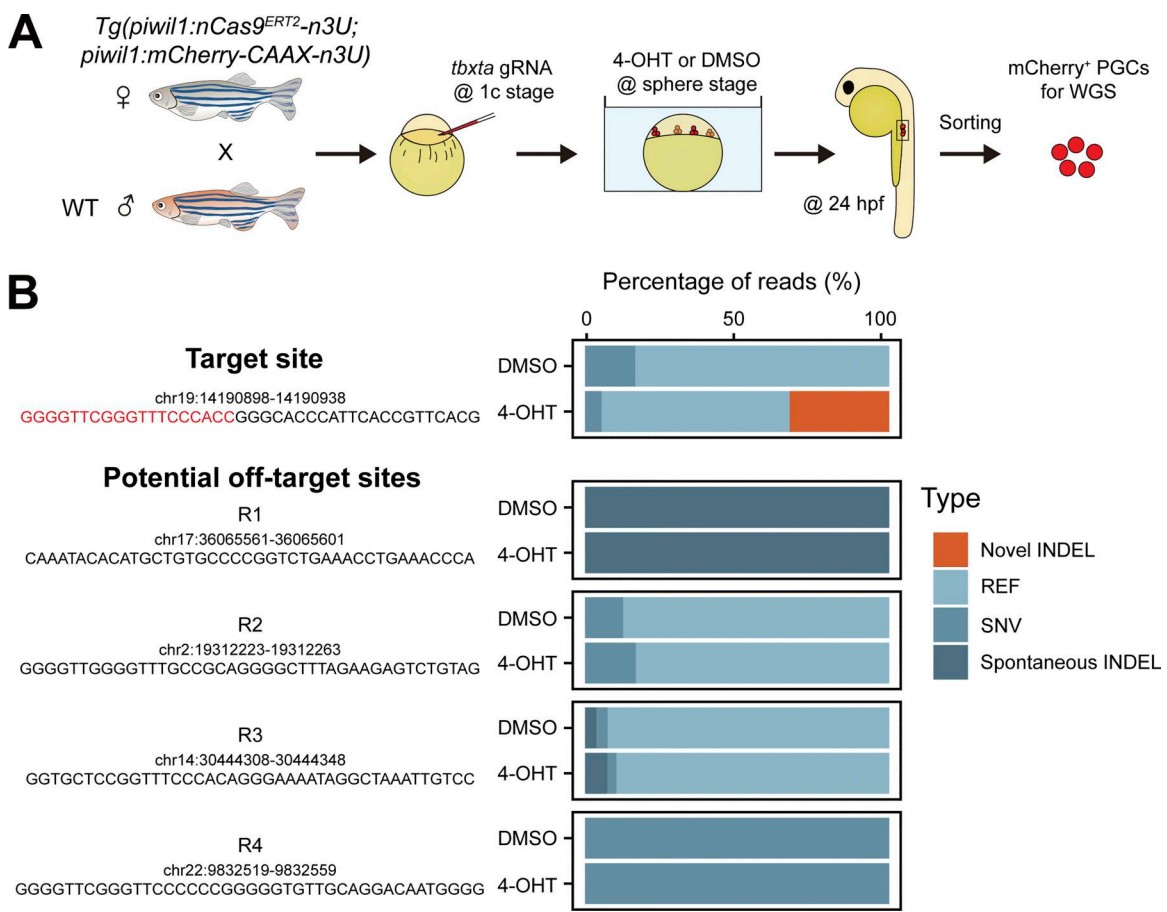

Figure S3.   **Off-target analysis of the *tbxta* locus with WGS and NGS. (A)** Schematic showing PGC collection for WGS. **(B)** Bar chart showing the mutation rates of PGCs within 4-OHT– or DMSO-treated embryos at the *tbxta* locus. Red letters indicate the sequence of *tbxta* target sites. Target site, *tbxta* gRNA target site; R1 to R4, the predicted *tbxta* gRNA off-target sites. INDEL, insertion-deletion; REF, reference genome; SNV, single-nucleotide variant.

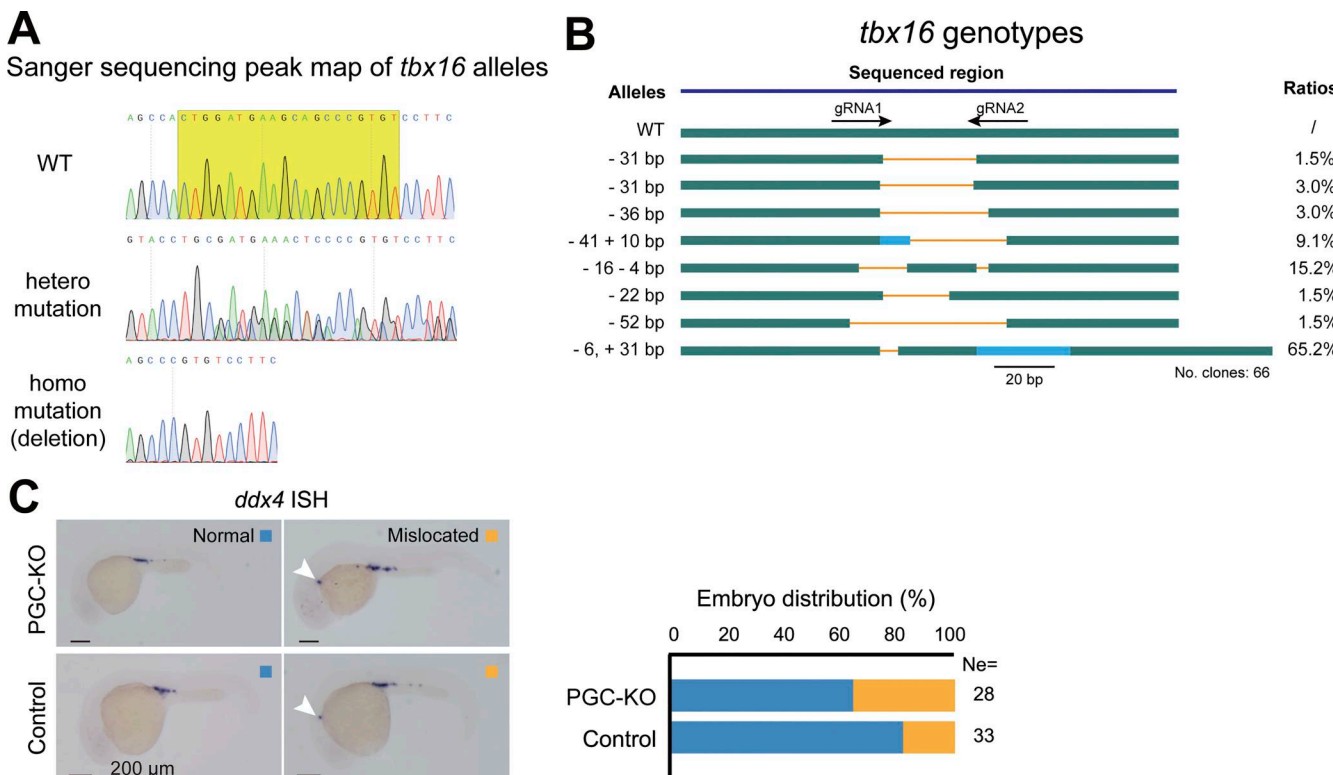

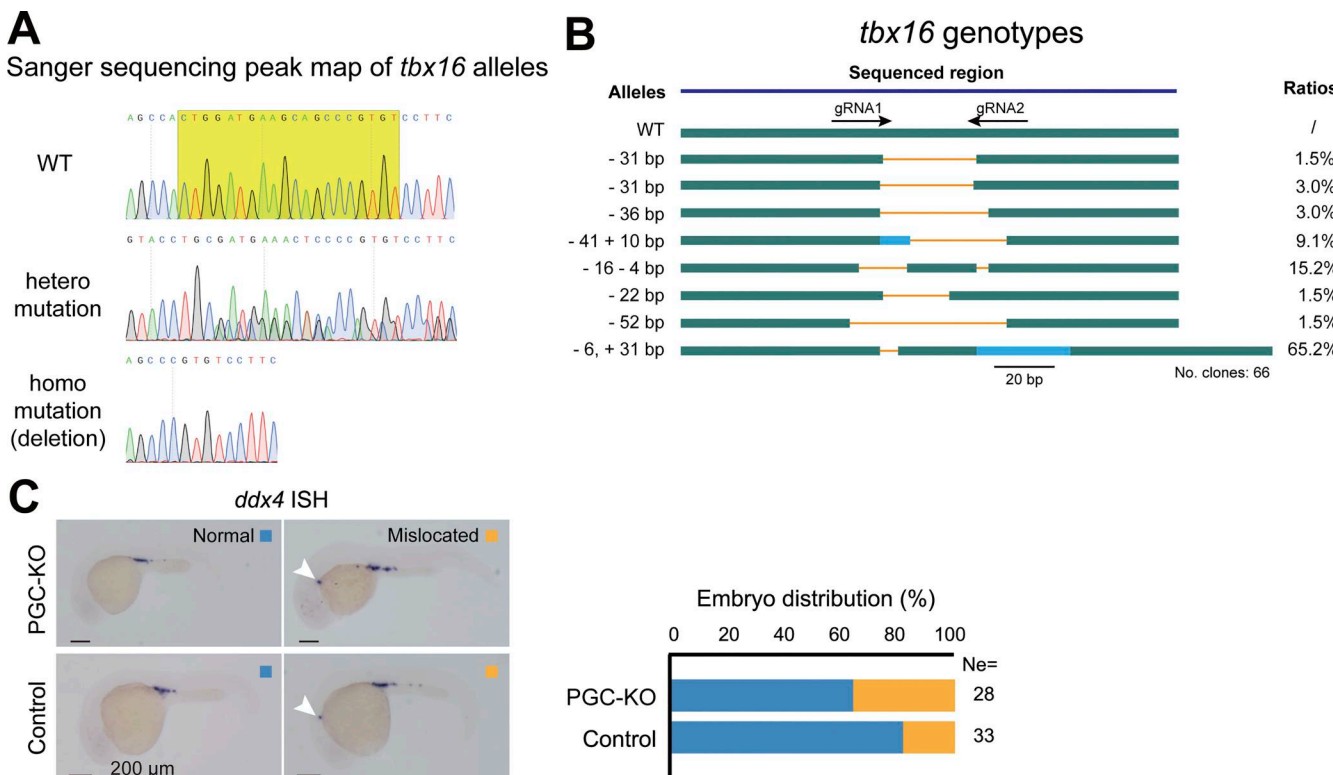

**A**

Sanger sequencing peak map of *tbx16* alleles

**B** *tbx16* genotypes

**C** *ddx4* ISH

Figure S4. **Mutation efficiency of *tbx16* alleles in PGCs. (A)** Sanger sequencing peak maps showing different mutation types of *tbx16* alleles (WT, heterozygous mutation, and homozygous mutation). The shadow yellow box indicates the *tbx16* target site. **(B)** Allelic mutant types and ratios. Green, WT sequence; gray, UTRs; orange line, deleted region; blue, inserted region. **(C)** ISH-detected *ddx4* expression in control and *tbx16* PGC-KO embryos at 24 hpf. The box plot showing the ratios of normal and mislocated PGCs in control and PGC-KO embryos. Ne, number of embryos. Source data are available for this figure: SourceData FS4.

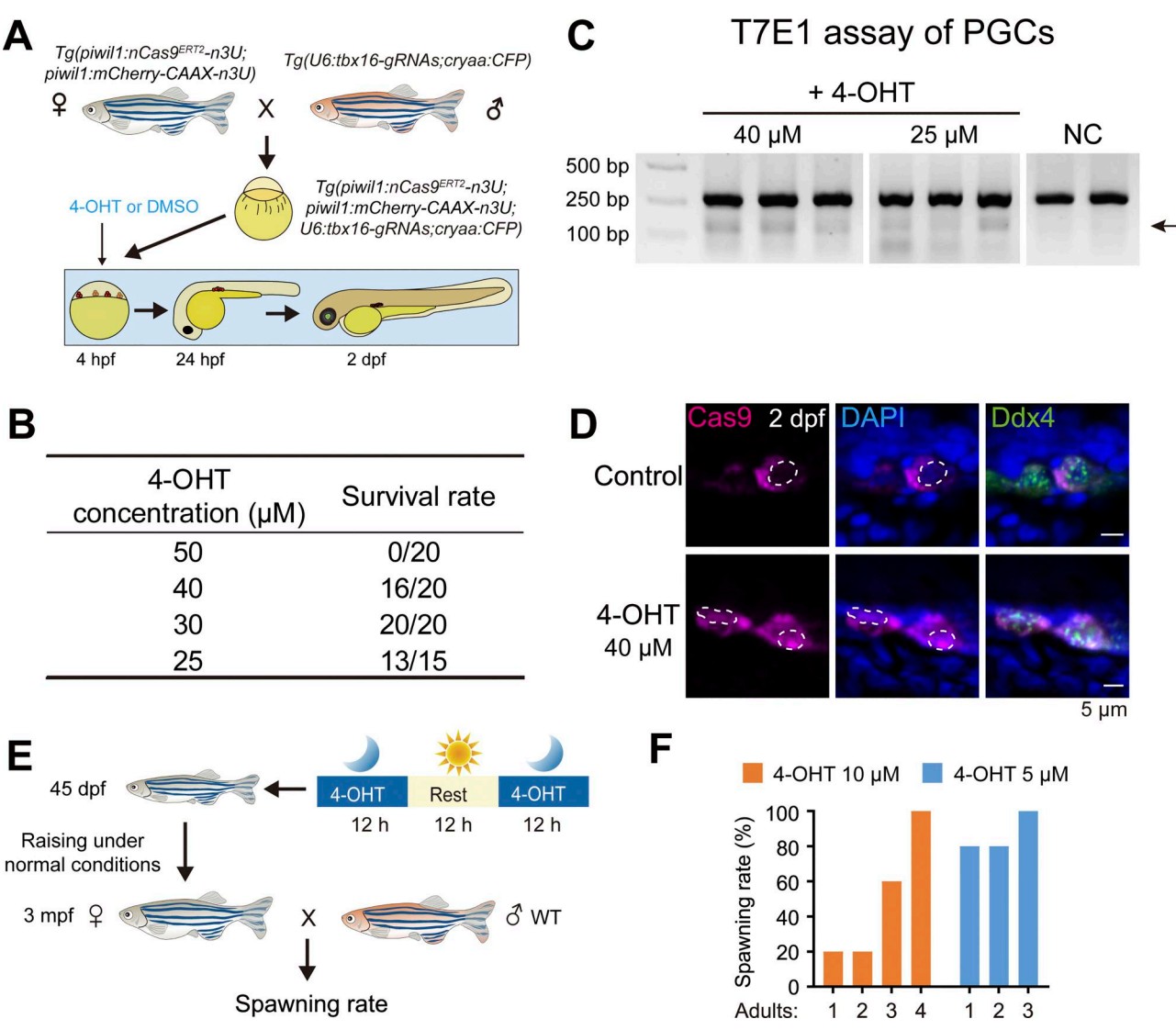

Figure S5. **4-OHT dosing selection for embryos and adults. (A)** Schematic showing the 4-OHT treatment strategy on embryos. **(B)** Table showing survival rates of embryos treated with different concentrations of 4-OHT from the sphere stage to 2 dpf. **(C)** Gel electrophoretic image showing the T7E1 assay outcome of the *tbx16* locus. The mutant band was indicated by the arrow. NC, negative control. **(D)** Immunostaining of Cas9 (magenta) and nuclei stained by DAPI (blue) in PGCs within the DMSO or 40 μM 4-OHT–treated embryos at 2 dpf. **(E)** Schematic showing the 4-OHT treatment strategy on 45 dpf females and detection of spawning rates. **(F)** Bar chart showing the spawning rates of adults treated with 10 and 5 μM 4-OHT at 45 dpf. Adults: The serial number of treated transgenic zebrafish. Source data are available for this figure: SourceData FS5.

**Provided online are Table S1, Table S2, and Table S3. Table S1 shows plasmid information. Table S2 shows primer information. Table S3 shows monoclonal Sanger sequencing results of *hwa* alleles in embryos.**

