## [Peer Review File · The Journal of Cell Biology]

Stage- and tissue-specific gene editing using 4-OHT inducible Cas9 in whole organism

Yaqi Li, Weiyang Zhang, Zihang Wei, Han Li, Xin Liu, Tao Zheng, Tursunjan Aziz, Cencan Xing, Anming Meng, and Xiaotong Wu

Corresponding Author(s): Xiaotong Wu, Tsinghua University

Review Timeline:

Submission Date:	2024-12-28
Editorial Decision:	2025-03-05
Revision Received:	2025-09-21
Editorial Decision:	2025-11-03
Revision Received:	2025-11-14
Editorial Decision:	2025-12-01
Revision Received:	2025-12-10

Monitoring Editor: Anna Huttenlocher

Scientific Editor: Dan Simon

Transaction Report:

DOI: <https://doi.org/10.1083/jcb.202412216>

March 5, 2025

Re: JCB manuscript #202412216

Xiaotong Wu
Tsinghua University

Dear Dr. Wu,

Thank you for submitting your manuscript entitled "Stage- and tissue-specific gene editing using 4-OHT inducible Cas9 in whole organism." Your manuscript has been assessed by expert reviewers, whose comments are appended below. Although the reviewers express potential interest in this work, significant concerns unfortunately preclude publication of the current version of the manuscript in JCB.

You will see that the Reviewers find your new method interesting and potentially important for the zebrafish community but they also express concerns about reproducibility and editing efficiency. Both request a more quantitative analysis of the nuclear translocation of Cas9 under 4-OHT treatment. Additionally, Reviewer #1 asks to show the off-target editing frequency and to add controls for the leaky editing assays while Reviewer #2 asks to clarify how the analyses of mutation efficiency were done and also for additional experiments to assess the editing efficiency. Fully addressing these and the other reviewer comments would be essential for a revised manuscript.

Please let us know if you are able to address the major issues outlined above and wish to submit a revised manuscript to JCB. Note that a substantial amount of additional experimental data likely would be needed to satisfactorily address the concerns of the reviewers. The typical timeframe for revisions is three to four months. If you anticipate any difficulties in meeting this aforementioned revision time limit, please contact us and we can work with you to find an appropriate time frame for resubmission. Please note that papers are generally considered through only one revision cycle, so any revised manuscript will likely be either accepted or rejected.

If you choose to revise and resubmit your manuscript, please also attend to the following editorial points. Please direct any editorial questions to the journal office.

GENERAL GUIDELINES:

Text limits: Character count is < 40,000, not including spaces. Count includes title page, abstract, introduction, results, discussion, and acknowledgments. Count does not include materials and methods, figure legends, references, tables, or supplemental legends.

Figures: Your manuscript may have up to 10 main text figures. To avoid delays in production, figures must be prepared according to the policies outlined in our Instructions to Authors, under Data Presentation, <https://jcb.rupress.org/site/misc/ifora.xhtml>. All figures in accepted manuscripts will be screened prior to publication.

*****IMPORTANT:** It is JCB policy that if requested, original data images must be made available. Failure to provide original images upon request will result in unavoidable delays in publication. Please ensure that you have access to all original microscopy and blot data images before submitting your revision. *******

Supplemental information: There are strict limits on the allowable amount of supplemental data. Your manuscript may have up to 5 supplemental figures. Up to 10 supplemental videos or flash animations are allowed. A summary of all supplemental material should appear at the end of the Materials and methods section.

Please note that JCB now requires authors to submit Source Data used to generate figures containing gels and Western blots with all revised manuscripts. This Source Data consists of fully uncropped and unprocessed images for each gel/blot displayed in the main and supplemental figures. Since your paper includes cropped gel and/or blot images, please be sure to provide one Source Data file for each figure that contains gels and/or blots along with your revised manuscript files. File names for Source Data figures should be alphanumeric without any spaces or special characters (i.e., SourceDataF#, where F# refers to the associated main figure number or SourceDataFS# for those associated with Supplementary figures). The lanes of the gels/blots should be labeled as they are in the associated figure, the place where cropping was applied should be marked (with a box), and molecular weight/size standards should be labeled wherever possible. Source Data files will be made available to reviewers during evaluation of revised manuscripts and, if your paper is eventually published in JCB, the files will be directly linked to specific figures in the published article.

Source Data Figures should be provided as individual PDF files (one file per figure). Authors should endeavor to retain a minimum resolution of 300 dpi or pixels per inch. Please review our instructions for export from Photoshop, Illustrator, and

PowerPoint here: <https://rupress.org/jcb/pages/submission-guidelines#revised>

If you choose to resubmit, please include a cover letter addressing the reviewers' comments point by point. Please also highlight all changes in the text of the manuscript.

Regardless of how you choose to proceed, we hope that the comments below will prove constructive as your work progresses. We would be happy to discuss them further once you've had a chance to consider the points raised. You can contact the journal office with any questions at cellbio@rockefeller.edu.

Thank you for thinking of JCB as an appropriate place to publish your work.

Sincerely,

Anna Huttenlocher, MD, PhD
Monitoring Editor
Journal of Cell Biology

Dan Simon, PhD
Scientific Editor
Journal of Cell Biology

Reviewer #1 (Comments to the Authors (Required)):

Stage- and tissue-specific gene editing using 4-OHT inducible Cas9 in whole organism
Li et al. J Cell Biol. 2025.

Summary: The manuscript by Li et al. describes and characterizes a new inducible system for conditional gene knockout, leveraging tissue-specific expression of a Cas9-ERT2 fusion protein. After testing various construct designs, the authors demonstrate that one in particular, nCas9ERT2, shows robust nuclear localization upon 4-OHT treatment and is capable of editing target genes in HEK293T in vitro experiments and in zebrafish embryos, where primordial germ cell (PGC) fates were examined. The authors provide solid evidence for the potential of this system to complement alternative approaches, and this manuscript is likely to be of broad interest to readers in disease modeling. Several points related to reproducibility and editing efficiency/fidelity need to be addressed. However, we recommend this manuscript be accepted pending revision that includes adequate responses to the concerns detailed below.

Major:

1. The result of the Cas9 nuclear localization experiment (figures 2B-C) shows only one replicate; additional replicates should be analyzed to show that this degree and stage of Cas9 localization is robust, and a broader field of view would show that the cells in focus are representative. Additionally, figure 2C needs to be referenced within the main text.
2. The possibility of off-target editing was not discussed or examined. Given this is a novel editing system, it is important to show and discuss the degree of off-target editing observed by nCas9ERT2 in 4-OHT-treated embryos. This would provide especially important context for the experiments relying on evaluation of developmental defects.
3. Leaky editing (i.e. editing in the absence of 4-OHT) is a concern that the authors address with a T7EI assay and a DMSO control group in the experiments. However, the DMSO group in the experiment shown in figure 3A is a much smaller sample size (less than half) compared to the 4-OHT group, and there is no mention of whether any editing was observed in the DMSO group in the experiment shown in figure 5B. Additionally, there was no DMSO group in the experiment shown in figure 4B. Justification or additional data is warranted here.

Minor:

1. In the adult induction experiment (figures 4B-C), the ratio of ventralized embryos was low (1.2-8%), which the authors ascribe to suboptimal 4-OHT treatments. Given the very high utility of an adult inducible system, it would be good for the authors to provide a comment on how the 4-OHT dosing schedule was chosen, and if possible, to include any additional data from alternate 4-OHT treatment schedules (potentially as a supplementary figure).
2. From the *tbx16* knockout experiments in PGCs, the conclusion that *tbx16* expression in PGCs is necessary for normal migration is justified, but the authors should elaborate on the possible role of somatic cells (SCs) in the normal migration of PGCs during somitogenesis since their involvement cannot be excluded, citing previous work where appropriate.
3. In figure 3B, the ratios for class I and class III are flipped compared to the main text. This should be corrected.

Reviewer #2 (Comments to the Authors (Required)):

This manuscript will be of interest to diverse researchers who seek to employ inducible knock-out tools in animal studies. In particular, this approach has the potential to advance zebrafish community where conditional loss-of-function tools remain challenging. The authors engineered cas9 with ERT2, enabling tamoxifen or 4-OHT treatment to translocate Cas9-ERT2 into the nucleus, thereby inducing target site cleavage and generating indel mutations. Similar approaches have been proposed in other studies, including Oakes et al. (Nat. Biotech., 2016), Liu et al. (Nat. Chem. Biol., 2016), Zhao et al. (Mol Ther Nucleic Acids, 2018), Lu et al. (Nucleic Acids Research, 2018), but to my knowledge this approach has not been examined in zebrafish.

This study aims to induce mutations in germ cells, a particularly challenging context for loss-of-function studies due to maternally derived mRNAs. First, the authors tested various combinations of cas9, NLS, and ERT2 and identified that Cas9-NLS-ERT2-NLS is successfully changed their location from cytosol to nucleus upon 4-OHT treatment in cell lines. By pairing with the piwil1 promoter, a primordial germ cell (PGC)-specific regulatory element, they generated a stable zebrafish line and examined gene disruption in germ cells of two distinct stages, including early embryogenesis and adult gonad. Furthermore, the authors expand this tool to mouse embryos. They tested three distinct loci in zebrafish and one locus in mouse.

While the manuscript presents an important and valuable strategy of conditional loss-of-function strategies, major key claims remain preliminary. Additional experiments are necessary to claim the efficiency of their tools. Moreover, the methods and statistical analyses are insufficiently applied, requiring further validation and clarification before publication.

Major comments

1. Efficiency of Cas9-ERT2 to target site disruption

Given that this manuscript introduces an important genetic tool, quantitative analysis data should be essential. Throughout the manuscript, I have concerns about whether Cas9-ERT2 efficiently and effectively induces target site disruption. Furthermore, I wonder what optimization has been examined by the authors. The following experiments should enhance the significance of this genetic tool and provide valuable guidance for readers interested in this technique.

1) Translocation efficiency of Cas9-ERT2

While the authors provide compelling images that 4-OHT treatment induces nuclear translocation by 4 hours post-treatment in early embryos (Fig. 2), there is no quantitative analysis. Fig. 2 provides one representative image for each stage without quantifying a ratio of Cas9 between cytosol and nucleus across a sufficient number of embryos. Also, there is no translocation analysis for adult gonads. Further quantitative analysis should be required.

How many Cas9+ PGCs are detectable per embryo and in gonad? Previous PGC studies and Fig. 1C suggested that multiple PGCs are labelled by the piwil1 promoter during early embryogenesis. However, Fig. 1D and 2B show a few cells (up to 4 cells in 1D at 24 hpf), raising concerns about whether all PGCs of piwil1:nCas9ERT2-n3U express Cas9-ERT2. It would be important to count how many Ddx4+ PGCs are detectable per embryo and how many Cas9+ cells are present in Ddx4+ PGC and Ddx4-PGC. Also, how many cells exhibit nuclear translocation of Cas9 upon 4-OHT treatment. The authors require to quantify it across a statistically significant number of embryos.

2) Cas9-ERT2 efficiency upon 4-OHT treatment

A key question is how efficiently 4-OHT treatment induces disruptive mutations. Fig 3B indicates 7/104 embryos exhibit severe defects when 4-OHT is treated in adult females. (Note that the text indicates 30/104, but figure shows 7/104. One would be typo). This efficiency (~7%) appears low. The authors evaluated indel mutations using sorted PGCs followed by subcloning and Sanger sequencing. This analysis indicate ~40% indel in PGC cells of piwil1:nCas9ERT2-n3U. However, it is unclear how many trials were performed. Moreover, sequencing individual construct from E.coli clones may not accurately determine the mutation rate. A more rigorous and appropriate method is to generate libraries and perform next generation sequences (NGS) to assess a massive number of genomes. Also, a ratio indicates in Fig 3D does not match to that of Fig 3C. The sum of values in Fig 3D is 100%, indicating that these ratios in Fig 3D may be calculated using mutated sequences, excluding wild-type results. This approach does not provide an accurate mutation efficiency.

Fig. 4 C also indicates only up to 8.6% embryos showed phenotypes after 4-OHT treatment in adults. A total tested animal number is three with 1.71 ~ 8.6% mutant ratios. This ratio is also unlikely efficient. Testing only three animals is insufficient as well. Wild-type control data are missing from the graph. Same as Fig. 3D, the sum of ratios of Fig. 4D is 100%, suggesting that wild-type sequence results may be excluded. In this case, indel efficiency cannot be determined.

Fig. 5D calculates the genotype of each PGC presumably using subcloning into a vector and sequencing several clones. But, the authors do not provide how many clones were tested to conclude wild-type, hetero, and homo mutants. Fig. 5E also appears to use only mutant sequence data excluding wild-type clones. For Fig. 5G, the authors likely used a paired t-test, but it is unclear why the author used a paired t-test but not an unpaired one. Only two cases exhibit high PGC mislocalizations for 4-OHT treatment. Thus, I assume that the unpaired t-test may not yield significant results. The authors require to provide justification of this statistical analysis.

Overall, the study does not provide clear evidence of the tool's efficiency in generating mutations.

The authors require to do following:

1) Include proper controls for Fig. 4-6.

The authors should include several positive and negative controls to accurately calculate the efficiency. The authors have the *piwi1:nCas9-n3U* strain. *piwi1:nCas9-n3U;sgRNA* double transgenic fish can serve as a positive control. *piwi1:nCas9ERT2-n3U* without sgRNA can serve as a negative control. Vehicle treatment with *piwi1:nCas9ERT2-n3U; sgRNA* double transgenic fish can serve as a negative control. Also please perform NGS-based mutation analysis instead of relying on subcloning.

2) Somatic cells

The authors require to use somatic or non-PGC cells to determine spatial *piwi1:nCas9ERT2-n3U* activity for Fig. 4 and 5.

2) Use reporter gene to accurately examine cas9 function

A standard approach to evaluate the efficiency of genome editing is to utilize EGFP or mCherry reporter. The authors require to use an EGFP reporter line that expresses fluorescence expression in PGC and/or non-PGC cells for their testing. Injection sgRNA rather than using a sgRNA stable line would be enough as generating a stable line does not make sense for revision. The authors require to test only PGC reporter expression is lost but non-PGC reporter expression retains. The authors already generated *piwi1:EGFP* line, which may serve as a PGC reporter.

2. Optimization

The authors used 40uM for early embryos and 5uM 4-OHT for adults. To my knowledge, 40uM is likely higher than standard 4-OHT concentrations used in zebrafish (up to 5-10uM) for other studies. The authors require to provide how efficiency is changed in different 4-OHT concentrations. If the authors already tested other 4-OHT conditions, please provide those optimization data. This data should be necessary for comprehensive analysis of their technique.

Minor

1. Fig. 2 labeling: Adding time points, such as 1,2,4, and 6 hpt (hours post-treatment) would be more informative than 30% Epi, Shield, 75% Epi, and Bud.
2. Line 161 *ea:EGFP* -> may typo, it could be *ef1a:EGFP*

AUTHORS' RESPONSES TO THE REVIEW

GENERAL RESPONSES

We deeply appreciate the reviewers for their valuable comments and suggestions, which have led to significant improvement of the manuscript. Here we include a point-by-point response letter to address the issues raised in our previous submission. Please note that we used Fig. 1, 2, 3, etc. to refer to figures in the manuscript and Fig. R1, R2, R3, etc. to refer to figures in this letter.

For the Reviewers' convenience, we also included a version of the manuscript in which the revised sections related to our responses to the Reviewers' comments are highlighted in blue.

Reviewer: 1

The manuscript by Li et al. describes and characterizes a new inducible system for conditional gene knockout, leveraging tissue-specific expression of a Cas9-ERT2 fusion protein. After testing various construct designs, the authors demonstrate that one in particular, nCas9ERT2, shows robust nuclear localization upon 4-OHT treatment and is capable of editing target genes in HEK293T in vitro experiments and in zebrafish embryos, where primordial germ cell (PGC) fates were examined. The authors provide solid evidence for the potential of this system to complement alternative approaches, and this manuscript is likely to be of broad interest to readers in disease modeling. Several points related to reproducibility and editing efficiency/fidelity need to be addressed. However, we recommend this manuscript be accepted pending revision that includes adequate responses to the concerns detailed below.

Responses: We thank the Reviewer for the summary of our findings. We have revised our paper according to the Reviewer's suggestions.

Major comments:

1. The result of the Cas9 nuclear localization experiment (figures 2B-C) shows only one replicate; additional replicates should be analyzed to show that this degree and stage of Cas9 localization is robust, and a broader field of view would show that the cells in focus are representative. Additionally, figure 2C needs to be referenced within the main text.

Responses: We thank the Reviewer for the excellent suggestion. We obtained a broader view of the whole embryos showing more PGCs within embryos expressing Cas9, which was also added to the revised Fig. 2B (Fig. R1A). Additionally, we measured the fluorescence intensity of Cas9 signals in the nucleus and throughout the whole PGC in more PGCs, and the statistical results are shown in the revised Fig. 2C (Fig. R1B), illustrating that 4-OHT treatment at the shield stage significantly increased the amount of nCas9^{ERT2} protein in the nucleus. Together, these data support the robustness of nCas9^{ERT2} nuclear translocation after 4-OHT treatment. Meanwhile, we added the index of Fig. 2C in the revised main text on page 6, lines 132-135: "Analysis of the Cas9 signal intensity ratio (nucleus to whole cell) revealed that, starting from

the shield stage, 4-OHT treated embryos exhibited significantly higher nucleus-to-whole-cell ratios of nCas9^{ERT2} protein in PGCs compared to control embryos (Fig. 2C).”

Figure R1. Nuclear translocation of nCas9^{ERT2} in *Tg(piwil1:nCas9^{ERT2}-n3U)* embryos. A, Immunostaining results of Cas9 (red) and Ddx4 (green) with nuclei stained by DAPI (blue) in control and 4-OHT-treated embryos from sphere stage to 30% epiboly, shield, 75% epiboly, and bud stages. Upper, a broader view of embryos. Scale bar, 20 μ m. Lower, the enlarged view of PGCs. Scale bar, 5 μ m. **B,** Box plots showing the ratio of Cas9 (red) signal intensity in the nucleus to whole PGCs. Nu, nucleus. No. PGCs, the number of PGCs. hpt, hours post treatment. *P*-values of *t*-test (unpaired, two-sided) were also labeled.

2. The possibility of off-target editing was not discussed or examined. Given this is a novel editing system, it is important to show and discuss the degree of off-target editing observed by nCas9^{ERT2} in 4-OHT-treated embryos. This would provide especially important context for the experiments relying on evaluation of developmental defects.

Responses: We thank the Reviewer for pointing this out. To check the off-target effects in our system, we performed whole genome sequencing (WGS) of PGCs in *Tg(piwil1:nCas9^{ERT2}-n3U;piwil1:mCherry-CAAX-n3U;U6:tbx16-gRNAs;cryaa:CFP)* triple transgenic embryos treated with 4-OHT or DMSO as a negative control from the sphere stage. Meanwhile, we established wild-type (WT) embryos co-injected with exogenous Cas9 protein and *tbx16* gRNAs as the positive control group (Fig. R2A). We collected about 200 PGCs sorted from 24 hpf embryos in the above treatments, and conducted whole genome amplification (WGA) of

each group of PGCs (Fig. R2A). Then, we performed WGS with these PGCs' genomes, analyzed the mutations at the *tbx16* gRNA target sites and the predicted potential off-target sites across the whole genome using Cas-OFFinder (Bae et al., 2014). The results showed that only two target sites of 4-OHT-treated triple transgenic embryos or Cas9 and gRNA-injected embryos obtained novel indel mutations with the efficiency of about 50%, while the predicted potential off-target sites exhibited single-nucleotide variant (SNV) or spontaneous indel, which are not caused by Cas9 mutagenesis (Fig. R2B). These results illustrate the low off-target editing effects of this genome editing system.

Figure R2. Off-target analysis of the *tbx16* locus with WGS. **A**, Schematic showing PGC collection and WGS. **B**, Bar chart showing the mutation efficiency in *tbx16* target sites and predicted potential off-targeting sites through the whole genome, in PGCs within 4-OHT or DMSO treated *Tg(piwil1:nCas9^{ERT2}-n3U; piwil1:mCherry-CAAX-n3U; U6:tbx16-gRNAs; cryaa:CFP)* embryos (left, inducible editing system) and within Cas9 injected or WT embryos (right, exogenous Cas9 protein). Red letters indicate the sequence of *tbx16* target sites. Target, *tbx16* gRNA target sites; R1 to R9, the predicted *tbx16* gRNA off-target sites by Cas-OFFinder (Bae et al., 2014). INDEL, insertion-deletion; REF, reference genome; SNV, single-nucleotide variant.

Similarly, we also checked the off-target effects of the *tbxta* target site. *Tg(piwill1:nCas9^{ERT2}-n3U;piwill1:mCherry-CAAX-n3U)* embryos were injected with *tbxta* gRNA at the 1-cell stage and treated with 4-OHT or DMSO from the sphere stage. We sorted about 200 PGCs and somatic cells (SCs) from treated embryos at the 24 hpf, respectively, and conducted WGA of pooled PGCs or somatic cells (Fig. R3A). Then, we performed WGS with 4-OHT-treated and control PGCs' genomes, analyzed the mutations at the *tbxta* gRNA target site and the off-target sites. The results showed that only the *tbxta* target site of 4-OHT-treated embryos contained novel indels, which further confirms the low off-target editing effects of our system (Fig. R3B). These results were added to the revised Fig. S3A-B.

Figure R3. Off-target analysis of *tbxta* locus with WGS and NGS. **A**, Schematic showing PGC collection and NGS. **B**, Bar chart showing the mutation efficiency in *tbxta* target sites and potential off-targeting sites through the whole genome, in PGCs within 4-OHT or DMSO-treated *Tg(piwill1:nCas9^{ERT2}-n3U;piwill1:mCherry-CAAX-n3U)* embryos. Red letters indicate the sequence of *tbxta* target sites. Target, *tbxta* gRNA target site; R1 to R4, the predicted *tbxta* gRNA off-target sites. INDEL, insertion-deletion; REF, reference genome; SNV, single-nucleotide variant.

3. Leaky editing (i.e. editing in the absence of 4-OHT) is a concern that the authors address with a T7EI assay and a DMSO control group in the experiments. However, the DMSO group in the experiment shown in figure 3A is a much smaller sample size (less than half) compared to the 4-OHT group, and there is no mention of whether any editing was observed in the DMSO group in the experiment shown in figure 5B. Additionally, there was no DMSO group in the experiment shown in figure 4B. Justification or additional data is warranted here.

Responses: We thank the Reviewer for this suggestion. The detailed analysis of control groups in various treatment conditions is quite necessary. Thus, we performed the following detections and analyses and added all these results to the corresponding figures in the revised figures.

For Fig. 3, we performed NGS to analyze the *tbxta* mutations in PGCs and somatic cells of 4-OHT or DMSO-treated embryos at 24 hpf to determine whether there was leaky editing in somatic cells treated with 4-OHT or PGCs and somatic cells from DMSO-treated embryos (Fig. R4A). We amplified the *tbxta* locus in PGCs and SCs from 4-OHT-treated and control embryos by nested PCR followed by next-generation sequencing (NGS) to determine *tbxta* mutation efficiency and specificity for PGCs. The sequencing results showed about a 40% mutation rate of the *tbxta* locus in PGCs from 4-OHT-treated embryos, but no reads containing *tbxta* mutations in PGCs and SCs from DMSO-treated embryos, as well as the SCs from 4-OHT-treated embryos (Fig. R4B). These results indicate no leaky editing in our system. These results are added to the revised Fig. 3D.

Figure R4. Leaky editing analysis of *tbxta* locus with NGS. A, Schematic showing PGC and somatic cell collection and NGS. **B,** Bar chart showing the mutation rates of PGCs and SCs within 4-OHT or DMSO-treated embryos at the *tbxta* locus. No. reads, the number of aligned reads.

For Fig. 5B, we performed the genotyping of PGCs within 4-OHT and DMSO-treated transgenic embryos and found no mutations in the *tbx16* locus in the DMSO group (Fig. R5A-B). Furthermore, we also performed immunostaining of Tbx16 protein in 4-OHT and DMSO-treated embryos at the bud stage, and there was no significant reduction of Tbx16 protein in the DMSO group (Fig. R5C). These data indicate that DMSO treatment would not induce either *tbx16* locus mutation or decreased Tbx16 protein levels. These results are also shown in the revised Fig. 5C and E.

Figure R5. Mutation of *tbx16* in 4-OHT-treated embryos. **A**, Schematic showing the 4-OHT treatment in the mutation strategy targeting the *tbx16* locus. **B**, Left, images showing the morphology of WT, *tbx16* whole-body knockout mutant (*tbx16^{tsu-G115}*) embryos, and *Tg(piwil1;nCas9^{ERT2}-n3U;U6:tbx16-gRNAs;cryaa:CFP)* embryos treated with 4-OHT at the sphere stage (PGC-KO). Right, bar chart showing genotypes of PGCs and somatic cells (Soma) in DMSO and 4-OHT-treated groups at 24 hpf. No. clones, the number of mono-clones for Sanger sequencing. **C**, Left, Immunostaining results of Tbx16 (green) and Ddx4 (red) with nuclei stained by DAPI (blue) in 4-OHT or DMSO-treated embryos from the sphere stage. Scale bar, 10 μm. Right, box plot showing the statistical results of the ratio of Tbx16 signals to DAPI in PGCs and somatic cells in either 4-OHT or DMSO-treated embryos. *P*-values of *t*-test (unpaired, two-sided) were also labeled.

For Fig. 4B, we raised the *Tg(piwil1:nCas9^{ERT2}-n3U;U6:hwa-gRNAs;ef1a:GFP)* young females and treated them with 4-OHT or DMSO at 45 dpf (days post fertilization). After 3 months post fertilization (mpf), we mated these females with WT males and collected their offspring for phenotype and genotype detection (Fig. R6A). There were no ventralized embryos produced by DMSO-treated females (Fig. R6B-C) and no mutations in the *hwa* locus in these embryos (Fig. R6D). In addition, we collected oocytes from 4-OHT or DMSO-treated females after 1 day post treatment and performed immunostaining of Cas9 (Fig. R6E). Notably, there were no Cas9 signals in nuclei of oocytes within DMSO-treated females (Fig. R6E-F). In sum, these data demonstrate the low leaky editing in our system. These results are added to the revised Fig. 4.

Figure R6. Mutation of *hwa* in 4-OHT-treated females. **A**, Schematic showing the 4-OHT treatment and subsequent mating and embryonic analysis for phenotype and genotype detection of *huluwa* (*hwa*) mutation in oocytes. Moon, night; Sun, daytime. **B**, Morphology of representative embryos at 24 hpf that were produced by transgenic females treated with 4-OHT or DMSO at 45 dpf. V1, V2, type 1 and 2 of ventralized embryos. **C**, Bar chart showing the distribution of different morphological phenotypes of embryos in **C**. Females: the serial number of treated transgenic females. No. embryos, the number of embryos produced by the corresponding females. **D**, Left, schematic showing different mutated genotypes of the *hwa* allele in the ventralized embryos. The light green boxes indicated the UTR of *hwa*, and the dark green boxes indicated the exons of *hwa*. Red lines and blue rectangles represented deletions

and insertions, respectively. F1, R1, the forward and reverse genotyping primer of *hwa* target site. Right, bar chart showing the proportion of different mutated *hwa* genotypes. Ven, ventralized embryos produced by 4-OHT-treated females. Nor, normal embryos produced by 4-OHT-treated females. No. embryos, the number of embryos. E, Immunostaining of Cas9 (green) and nuclei stained by DAPI (blue) in ovaries within the 4-OHT or DMSO-treated females at 45 dpf. F, Box plots showing the ratio of Cas9 (red) signal intensity in the nucleus to the cytoplasm of PGCs (left), and the signal intensity ratio of Cas9 (red) to DAPI (blue) in the nucleus of PGCs (right). Nu, nucleus. Cyto, cytoplasm. No. cells, the number of oocytes. *P*-values of *t*-test (unpaired, two-sided) were also labeled.

Minor comments:

1. In the adult induction experiment (figures 4B-C), the ratio of ventralized embryos was low (1.2-8%), which the authors ascribe to suboptimal 4-OHT treatments. Given the very high utility of an adult inducible system, it would be good for the authors to provide a comment on how the 4-OHT dosing schedule was chosen, and if possible, to include any additional data from alternate 4-OHT treatment schedules (potentially as a supplementary figure).

Responses: We thank the Reviewer for pointing this out. We agree that providing the 4-OHT dosing schedule is indeed crucial for the application and promotion of our system.

Actually, we employed two types of 4-OHT dosing selection schemes for our system, tailored to the developmental stages.

For adult treatment, we performed a spawning rate detection by 5 or 10 μM 4-OHT treatment (Fig. R7A), and observed a severe disruption of reproduction in the 10 μM 4-OHT group (Fig. R7B). Thus, although the relatively lower editing efficiency, we still chose a concentration of 5 μM for adult treatment to ensure the spawning rate.

Next, for early embryogenesis, we tested four concentrations of 4-OHT: 25, 30, 40, and 50 μM from the sphere to the 2 dpf stages, and detected the survival rates of treated embryos at the 2 dpf, indicating a severe cytotoxic effect with the 50 μM treatment (Fig. R7C-D). In addition, by the T7EI assay, we found that the genome editing efficiency of 40 μM 4-OHT treatment was higher than that of 25 μM (Fig. R7E). Via immunostaining of Cas9 protein, we revealed that 40 μM 4-OHT treatment was sufficient to trigger nCas9^{ERT2} nuclear translocation in PGCs (Fig. R7F). Thus, a concentration of 40 μM was used for early embryos in the subsequent experiments in our manuscript. We added all these results to the revised supplementary figure 5 (Fig. S5) and also discussed it on pages 12-13, lines 321-334:

“In our inducible system, nCas9^{ERT2} nuclear translocation relies on the action of 4-OHT. The concentration and treatment duration of 4-OHT should be optimized on a case-by-case basis. Actually, we employed two types of 4-OHT dosing selection schemes for our system, tailored to the developmental stages. For early embryogenesis, we tested four concentrations of 4-OHT: 25, 30, 40, and 50 μM from the sphere to the 2 dpf stages, and detected the survival rates of treated embryos at the 2 dpf, indicating a severe cytotoxic effect with the 50 μM treatment (Fig. S5A-B). In addition, by the T7EI assay, we found that the genome editing efficiency of 40 μM 4-OHT treatment was higher than that of 25 μM (Fig. S5C). Via immunostaining of Cas9 protein, we revealed that 40 μM 4-OHT treatment was sufficient to

trigger $nCas9^{ERT2}$ nuclear translocation in PGCs (Fig. S5D). Thus, a concentration of 40 μM was used for early embryos in the subsequent experiments. Next, for adult treatment, we performed a spawning rate detection by 5 or 10 μM 4-OHT treatment (Fig. S5E), and observed a severe disruption of reproduction in the 10 μM 4-OHT group (Fig. S5F). Thus, although the relatively lower editing efficiency, we still chose a concentration of 5 μM for adult treatment to ensure the spawning rate.”

Figure R7. 4-OHT dosing selection for embryos and adults. **A**, Schematic showing the 4-OHT treatment with 45 dpf females and detection of spawning rates. **B**, Bar chart showing the spawning rates of adults treated with 10 μM or 5 μM 4-OHT at 45 dpf. Adults: the serial number of treated transgenic zebrafish. **C**, Schematic showing the 4-OHT treatment for embryos. **D**, Table showing survival rates of embryos treated with different concentrations of 4-OHT from the sphere stage to the 2 dpf stage. **E**, Gel electrophoretic image showing T7E1 assay outcome of the *tbx16* locus. The mutant band was indicated by the arrow. NC, negative control. **F**, Immunostaining of Cas9 (red) and nuclei stained by DAPI (blue) in PGCs within the DMSO or 40 μM 4-OHT-treated embryos at 2 dpf.

2. From the *tbx16* knockout experiments in PGCs, the conclusion that *tbx16* expression in PGCs is necessary for normal migration is justified, but the authors should elaborate on the possible role of somatic cells (SCs) in the normal migration of PGCs during somitogenesis since their involvement cannot be excluded, citing previous work where appropriate.

Responses: We thank the Reviewer for the wonderful suggestions. We have discussed the possible roles of somatic cells during PGC migration at somitogenesis stages in the Discussion section (Pages 13-14, Lines 341-368).

“Around the shield stage, four clusters of PGCs begin migrating toward the dorsal side of embryos, requiring chemokine signal guidance from somatic cells (Aguero et al., 2017; Raz, 2003). PGCs rely on the intracellular chemokine receptor *Cxcr4b* to respond to the chemokine *Cxcl12a* (also known as *Sdf1*) expressed by surrounding somatic cells, migrating directionally toward regions with a higher level of *Cxcl12a* (Boldajipour et al., 2008; Doitsidou et al., 2002; Paksa and Raz, 2015; Weidinger et al., 1999). Knockdown or knockout of *Cxcr4b* or *Cxcl12a* results in an obvious incorrect migration of PGCs (Doitsidou et al., 2002; Molyneaux et al., 2003). *Tbx16* has also been reported to participate in regulating the migration of PGCs. In 24 hpf *tbx16* zygotic mutant (*Ztbx16*) embryos, a portion of PGCs is ectopically localized in the head or tail region of embryos (Weidinger et al., 1999). The authors speculated that this phenotype is caused by a deficiency of *Tbx16* protein, which causes an aberrant expression pattern of *cxcl12a* in somatic cells and disrupts PGC migration. *Cxcl12a/b* and *Cxcr4a* are also involved in somatic cell migration during gastrulation (Mizoguchi et al., 2008). During zebrafish endodermal development, the *Tbx16* protein might directly act on transcription of *cxcl12a/b*, which is downregulated in the *Ztbx16* embryos at the 75% epiboly stage (Garnett et al., 2009; Nelson et al., 2017). Another study also reported that the expression pattern of *cxcl12a* is altered in *Ztbx16* embryos (Weidinger et al., 1999).

Considering that expression levels of *tbx16* in PGCs and SCs are basically the same throughout early embryogenesis (Skvortsova et al., 2019), whether *tbx16* expressed in PGCs participates in PGC development and migration remains to be explored. This work offers an alternate explanation for the abnormal PGC migration phenotype observed in *Ztbx16* embryos. With this inducible gene editing system, we tissue-specifically knocked out *tbx16* in PGCs to avoid potential interference from *tbx16* deficiency in SCs on PGC migration. We found that PGC-specific knockout of *tbx16* disrupted PGC migration, resulting in a significant increase in the proportion of embryos with ectopic PGCs. This result indicates that *tbx16* expressed in PGCs plays a cell-autonomous role in the PGC migration process, and its absence increases the likelihood of incorrect PGC migration.”

3. In figure 3B, the ratios for class I and class III are flipped compared to the main text. This should be corrected.

Responses: We thank the Reviewer for pointing this out and apologize for the typo in the figure. We have corrected it in the revised manuscript.

Reviewer: 2

This manuscript will be of interest to diverse researchers who seek to employ inducible knock-out tools in animal studies. In particular, this approach has the potential to advance zebrafish

community where conditional loss-of-function tools remain challenging. The authors engineered cas9 with ERT2, enabling tamoxifen or 4-OHT treatment to translocate Cas9-ERT2 into the nucleus, thereby inducing target site cleavage and generating indel mutations. Similar approaches have been proposed in other studies, including Oakes et al. (Nat. Biotech., 2016), Liu et al. (Nat. Chem. Biol., 2016), Zhao et al. (Mol Ther Nucleic Acids, 2018), Lu et al. (Nucleic Acids Research, 2018), but to my knowledge this approach has not been examined in zebrafish.

This study aims to induce mutations in germ cells, a particularly challenging context for loss-of-function studies due to maternally derived mRNAs. First, the authors tested various combinations of cas9, NLS, and ERT2 and identified that Cas9-NLS-ERT2-NLS is successfully changed their location from cytosol to nucleus upon 4-OHT treatment in cell lines. By pairing with the piwil1 promoter, a primordial germ cell (PGC)-specific regulatory element, they generated a stable zebrafish line and examined gene disruption in germ cells of two distinct stages, including early embryogenesis and adult gonad. Furthermore, the authors expand this tool to mouse embryos. They tested three distinct loci in zebrafish and one locus in mouse.

While the manuscript presents an important and valuable strategy of conditional loss-of-function strategies, major key claims remain preliminary. Additional experiments are necessary to claim the efficiency of their tools. Moreover, the methods and statistical analyses are insufficiently applied, requiring further validation and clarification before publication.

Responses: We thank the Reviewer for the excellent summary of our findings and for appreciating the value of our work. We have revised our paper according to the Reviewer's suggestions.

Major comments:

1. Efficiency of Cas9-ERT2 to target site disruption

Given that this manuscript introduces an important genetic tool, quantitative analysis data should be essential. Throughout the manuscript, I have concerns about whether Cas9-ERT2 efficiently and effectively induces target site disruption. Furthermore, I wonder what optimization has been examined by the authors. The following experiments should enhance the significance of this genetic tool and provide valuable guidance for readers interested in this technique.

Responses: We thank the Reviewer for pointing this out. We would like to provide an overview of the optimization strategy applied to our system before we show the results for the following suggested experiments one by one.

Specifically, to achieve spatiotemporal gene editing, we performed the following optimizations:

1, Through immunostaining of the Cas9 protein, we screened various combinations of Cas9, NLS, and ERT2, and identified that the nCas9^{ERT2} protein flanked by two different nuclear localization signals on either side would efficiently improve its translocation into the nucleus upon induction of 4-OHT (Fig. S1 and 1A). In addition, we validated its genome editing efficiency by T7EI assay (Fig. 1B).

2, Optimization of the germ-line specific promoter. Besides optimization of the *piwill* promoter, which has been shown in Fig. S2, we also tested *ca15b* and *tdrd7a* promoters by constructing transgenic lines and checked the reporter fluorescence protein expression. However, the expression of mCherry driven by these promoters displayed strong maternal expression, a ubiquitous expression pattern at early stages, and relatively weak PGC-specific expression till at the bud stage (Fig. R8A-B). Thus, these two promoters were not suitable for PGC editing during the early stages, and we did not include these results in our previous figures, nor do we add them in the revised version.

Figure R8. Reporter expression driven by *ca15b* and *tdrd7a* promoters. **A.** Fluorescence images showing the mCherry expression in a transgenic line with the 5.3-kb *ca15b* promoter. CAAX, cell membrane localizing signal. c3U, 3' untranslated region of the zebrafish *ca15b* gene. **B.** Fluorescence images showing the mCherry expression in a transgenic line with the 3.3-kb *tdrd7a* promoter. t3U, 3' untranslated region of the zebrafish *tdrd7a* gene. White arrowheads indicate PGCs.

3, We employed two types of 4-OHT dosing selection schemes for our system, tailored to the developmental stages. For early embryogenesis, we tested four concentrations of 4-OHT from

the sphere to the 2 dpf stages, and checked the survival rates, nCas9^{ERT2} nuclear translocation, and editing rates of treated embryos. For adult treatment, we performed a spawning rate detection using two concentrations of 4-OHT treatment. Then, we chose the 4-OHT concentrations, ensuring low cell toxicity and higher editing efficiency.

1-1. Translocation efficiency of Cas9-ERT2

While the authors provide compelling images that 4-OHT treatment induces nuclear translocation by 4 hours post-treatment in early embryos (Fig. 2), there is no quantitative analysis. Fig. 2 provides one representative image for each stage without quantifying a ratio of Cas9 between cytosol and nucleus across a sufficient number of embryos. Also, there is no translocation analysis for adult gonads. Further quantitative analysis should be required.

Responses: We thank the Reviewer for the excellent suggestions. We agree that the quantitative analysis of the nCas9^{ERT2} translocation efficiency is essential for demonstrating the feasibility and effectiveness of this inducible system.

For Fig. 2, we repeated this immunostaining experiment, measured the fluorescence intensity of Cas9 signals in the nucleus and whole PGCs, calculated the ratio of Cas9 signal intensity in the nucleus to total PGCs, and found that the efficiency of nuclear translocation of Cas9 increases from 0% (0 of 4) at 30% epiboly stage to 20% (1 of 5) at shield stage, then to 90.9% (10 of 11) at 75% epiboly, and 70% (7 of 10) at bud stages, respectively (Fig. R9A). These data illustrate that 4-OHT treatment could significantly increase the amount of nCas9^{ERT2} protein in the nucleus from the shield stage (two hours post-treatment) (Fig. R9A). We added these results to the revised figure (Fig. 2B-C).

For adult gonads, we collected ovaries from *Tg(piwill:nCas9^{ERT2}-n3U;U6:hwa-gRNAs;efla:GFP)* young females at 45 dpf treated with 4-OHT or DMSO. Then, we performed immunostaining of Cas9 in their oocytes after 1 day post-treatment and found that nCas9^{ERT2} in oocytes could be translocated into the nucleus after 4-OHT treatment but not in the DMSO-treated group (Fig. R9B), which could be supported by analysis of the Cas9 signal distribution in the cytosol and nucleus within oocytes (Fig. R9C). In sum, these data indicated that nCas9^{ERT2} nuclear translocation in early embryos and oocytes within adult ovaries after 4-OHT treatment is robust. We added these results to the revised figure (Fig. 4C-D).

Figure R9. Nuclear translocation of nCas9^{ERT2} in *Tg(piwil1:nCas9^{ERT2}-n3U)* embryos and oocytes. **A**, Box plots showing the ratio of Cas9 (red) signal intensity in the nucleus to total PGCs. Nu, nucleus. No. PGCs, the number of PGCs. *P*-values of *t*-test (unpaired, two-sided) were also labeled. **B**, Immunostaining of Cas9 (green) and nuclei stained by DAPI (blue) in ovaries within the 4-OHT or DMSO-treated females at 45 dpf. **C**, Box plots showing the ratio of Cas9 (red) signal intensity in the nucleus to the cytoplasm of PGCs (left), and the signal intensity ratio of Cas9 (red) to DAPI (blue) in the nucleus of PGCs (right). Nu, nucleus. Cyto, cytoplasm. No. cells, the number of oocytes. *P*-values of *t*-test (unpaired, two-sided) were also labeled.

How many Cas9+ PGCs are detectable per embryo and in gonad? Previous PGC studies and Fig. 1C suggested that multiple PGCs are labelled by the piwil1 promoter during early embryogenesis. However, Fig. 1D and 2B show a few cells (up to 4 cells in 1D at 24 hpf), raising concerns about whether all PGCs of piwil1:nCas9^{ERT2}-n3U express Cas9-ERT2. It would be important to count how many Ddx4+ PGCs are detectable per embryo and how many Cas9+ cells are present in Ddx4+ PGC and Ddx4- PGC. Also, how many cells exhibit nuclear translocation of Cas9 upon 4-OHT treatment. The authors require to quantify it across a statistically significant number of embryos.

Responses: We thank the Reviewer for their excellent suggestions. Validation of the PGC-specific expression of nCas9^{ERT2} is indeed necessary to demonstrate the feasibility and effectiveness of our inducible system.

For Fig. 1D, to enlarge the sample size for statistical analysis, we repeated the Ddx4 and Cas9 immunostaining experiment (Fig. R10A) and counted the number of Cas9-positive and -negative PGCs (Ddx4-positive, Ddx4-) in transgenic embryos at different developmental

stages. The analysis of the ratio of Cas9-positive PGCs to total PGCs revealed that more than 70% of PGCs from the sphere to 24 hpf expressed Cas9 (Fig. R10B), which indicates that the optimized 2.5 kb *piwill* promoter and *nanos1* 3'UTR are sufficient to drive the PGC-specific expression of nCas9^{ERT2}. We added these results to the revised figure (Fig. 1D-E).

Figure R10. PGC-specific expression of nCas9^{ERT2} in transgenic embryos. **A**, Immunostaining results of Cas9 (red) and Ddx4 (green) with nuclei stained by DAPI (blue) of *Tg(piwill:nCas9^{ERT2}-n3U)* embryos at the 512-cell, sphere, shield, and 24 hpf stages. Inserts showed enlarged areas harboring PGCs. **B**, Bar chart showing the ratio of Cas9-positive PGCs (Ddx4 and Cas9 double-positive cells) to total PGCs (Ddx4 positive cells) in *Tg(piwill:nCas9^{ERT2}-n3U)* embryos in **A**. No. PGCs, the number of PGCs.

In addition, similarly as previous comment, to figure out the efficiency of nuclear translocation of nCas9^{ERT2} upon 4-OHT treatment, we performed statistical analysis of ratio with nuclear translocated nCas9^{ERT2} PGCs to all PGCs and found that the efficiency of nuclear translocation of Cas9 increases from 0% (0 of 4) at 30% epiboly stage to 20% (1 of 5) at shield stage, then to 90.9% (10 of 11) at 75% epiboly, and 70% (7 of 10) at bud stages, respectively (Fig. R9A).

1-2. Cas9-ERT2 efficiency upon 4-HT treatment

A key question is how efficiently 4-OHT treatment induces disruptive mutations. Fig 3B indicates 7/104 embryos exhibit severe defects when 4-OHT is treated in adult females. (Note that the text indicates 30/104, but figure shows 7/104. One would be typo). This efficiency (~7%) appears low. The authors evaluated indel mutations using sorted PGCs followed by subcloning and Sanger sequencing. This analysis indicate ~40% indel in PGC cells of *piwill:nCas9ERT2-n3U*. However, it is unclear how many trials were performed. Moreover, sequencing individual construct from E.coli clones may not accurately determine the mutation rate. A more rigorous and appropriate method is to generate libraries and perform next generation sequences (NGS) to assess a massive number of genomes. Also, a ratio indicates in Fig 3D does not match to that of Fig 3C. The sum of values in Fig 3D is 100%, indicating that these ratios in Fig 3D may be calculated using mutated sequences, excluding wild-type results. This approach does not provide an accurate mutation efficiency.

Responses: We thank the Reviewer for pointing this out and **apologize for the typo in the figure**. There were 28.8% (30/104) embryos in Class III and 6.7% (7/104) of embryos in Class I in the *tbxta* gRNA-injected *Tg(piwill:nCas9-n3U;piwill:mCherry-CAAX-n3U)* embryos (Fig.

R11A-B). The Class II and Class III embryos had varying degrees of defective notochord development, which mimics *tbxta* whole-body knockout mutants (Wu et al., 2018). *tbxta* is zygotically expressed in mesendodermal progenitors and the notochord (Schulte-Merker et al., 1992), but not in PGCs. Thus, *tbxta* gRNA injection into *Tg(piwill:nCas9-n3U)* embryos, which obtained ubiquitous expression of Cas9 in both PGCs and somatic cells before the sphere stage, would trigger mutation of *tbxta* in somatic cells, consequently resulting in a high percentage (93.3%, 97/104, combined Class II and Class III) of embryos with notochord defects (Fig. R11A-B). These results indicate that, without temporal control of gene editing, non-specific gene editing in somatic cells in *Tg(piwill:nCas9-n3U)* embryos at early stages is hard to avoid. Thus, we need improve the specificity of PGC genome editing technology.

Using our nCas9^{ERT2} system, we repeated this assay to detect the morphology of DMSO- or 4-OHT-treated *tbxta* gRNA-injected *Tg(piwill:nCas9^{ERT2}-n3U;piwill:mCherry-CAAX-n3U)* embryos at 24 hpf. Both DMSO-treated and 4-OHT-treated embryos showed normal notochord development (Fig. R11A-B), suggesting that there were no or at least extremely low somatic cell mutations at the *tbxta* locus. These data support a higher specificity of our nCas9^{ERT2} system than the traditional non-temporally controlled *Tg(piwill:nCas9-n3U)* editing system.

To validate the genome editing specificity in the nCas9^{ERT2} system, we used Sanger sequencing of *tbxta* gRNA target sites and found a high somatic cell mutation rate in *Tg(piwill:nCas9-n3U)* editing system (Fig. R11C) in our previous manuscript. However, as suggested by the Reviewer, due to the limited number of clones, it was not rigorous to negate the existence of somatic cell mutations in the nCas9^{ERT2} system. Thus, we performed NGS with the *tbxta* gRNA target site of PGCs and somatic cells from the DMSO- or 4-OHT-treated nCas9^{ERT2} embryos. We analyzed at least 5 million reads for each sample, which is sufficient to determine whether a mutation is present. The sequencing results revealed an efficient mutation rate of the *tbxta* target site induced by nCas9^{ERT2} in PGCs within 4-OHT-treated embryos (Fig. R11D), which was comparable with Sanger sequencing results (Fig. R11C). Whereas there were no mutated reads in PGCs and SCs from the DMSO-treated embryos, and SCs from the 4-OHT-treated embryos (Fig. R11D). Therefore, these results further demonstrate the specificity and efficiency of our system. These figures were all added to the revised Fig. 3.

Figure R11. Specificity and editing efficiency of the *nCas9^{ERT2}* system. **A**, Schematics of experimental procedures. $Tg(piwil1:nCas9^{ERT2}-n3U)$ (left) or $Tg(piwil1:nCas9-n3U)$ (right) transgenic females were crossed to WT males, which produced embryos with maternal expression of inducible *nCas9^{ERT2}* or non-inducible *nCas9*, respectively. SCs, somatic cells. **B**, Left, Bar chart showing the ratio of different morphological types of *tbxta*-gRNA injected 4-OHT- or DMSO-treated *nCas9^{ERT2}* embryos, and *nCas9* embryos at 24 hpf. n, the number of embryos. Right, Morphology of representative embryos in each class. The boxed areas of notochord were enlarged. **C**, Mutation rate of the *tbxta* locus in mCherry-positive PGCs or mCherry-negative SCs in different groups at 24 hpf. A region in the *tbxta* locus was amplified and cloned, followed by sequencing individual clones. N, number of sequenced clones. **D**, Bar chart showing the mutation rate of the *tbxta* locus in mCherry-positive PGCs or mCherry-negative SCs in *tbxta*-gRNA injected 4-OHT- or DMSO-treated *nCas9^{ERT2}* embryos at 24 hpf, calculated with NGS results. No. reads, the number of aligned reads per target site.

By the way, the original Fig. 3D showed the representative mutation forms detected by Sanger sequencing, which have been removed in the revised Figures because we added the more accurate NGS results.

Fig. 4 C also indicates only up to 8.6% embryos showed phenotypes after 4-OHT treatment in adults. A total tested animal number is three with 1.71 ~ 8.6% mutant ratios. This ratio is also unlikely efficient. Testing only three animals is insufficient as well. Wild-type control data are missing from the graph. Same as Fig. 3D, the sum of ratios of Fig. 4D is 100%, suggesting that

wild-type sequence results may be excluded. In this case, indel efficiency cannot be determined.

Responses: We thank the Reviewer for pointing this out. We agree that the appropriate number of samples for statistical analysis is necessary for illustrating the effectiveness of this system.

We raised more *Tg(piwil1:nCas9^{ERT2}-n3U;U6:hwa-gRNAs;ef1a:GFP)* young females and treated them with 4-OHT or DMSO at 45 dpf. After 3 mpf, we mated an extra four females with WT males, respectively, and collected their offspring for phenotype and genotype detection (Fig. R6A, this figure index is consistent with the response to Reviewer 1#). We revealed that all seven 4-OHT-treated females could produce ventralized embryos, with the ratios ranging from 1.71% (2/117) to 14.8% (12/81), meanwhile, no ventralized embryos were produced by four DMSO-treated females (Fig. R6B-C). Further editing efficiency analysis of offspring from DMSO-treated females revealed that there were no mutations in the *hwa* locus detected (Fig. R6D, DMSO). Notably, all ventralized embryos from the cross of 4-OHT-treated females and WT males were heterozygotes with various mutation forms at the *hwa* locus (Fig. R6D). Even about 30% of normal embryos from the 4-OHT-treated females were heterozygotes. In sum, these data support that the genome in immature oocytes can be edited by our nCas9^{ERT2} system. These figures were all added to the revised Fig. 4.

Figure R6. Mutation of *hwa* in 4-OHT-treated females. **A**, Schematic showing the 4-OHT treatment and subsequent mating and embryonic analysis for phenotype and genotype detection of *huluwa* (*hwa*) mutation in oocytes. Moon, night; Sun, daytime. **B**, Morphology of representative embryos at 24 hpf that were produced by transgenic females treated with 4-OHT or DMSO at 45 dpf. V1, V2, type 1 and 2 of ventralized embryos. **C**, Bar chart showing the distribution of different morphological phenotypes of embryos in **C**. Females: the serial number of treated transgenic females. No. embryos, the number of embryos produced by the corresponding females. **D**, Left, schematic showing different mutated genotypes of the *hwa* allele in the ventralized embryos. The light green boxes indicated the UTR of *hwa*, and the dark green boxes indicated the exons of *hwa*. Red lines and blue rectangles represented deletions and insertions, respectively. F1, R1, the forward and reverse genotyping primer of *hwa* target site. Right, bar chart showing the proportion of different mutated *hwa* genotypes. Ven, ventralized embryos produced by 4-OHT-treated females. Nor, normal embryos produced by 4-OHT-treated females. No. embryos, the number of embryos.

Fig. 5D calculates the genotype of each PGC presumably using subcloning into a vector and sequencing several clones. But, the authors do not provide how many clones were tested to conclude wild-type, hetero, and homo mutants. Fig. 5E also appears to use only mutant

sequence data excluding wild-type clones. For Fig. 5G, the authors likely used a paired t-test, but it is unclear why the author used a paired t-test but not an unpaired one. Only two cases exhibit high PGC mislocalizations for 4-OHT treatment. Thus, I assume that the unpaired t-test may not yield significant results. The authors require to provide justification of this statistical analysis.

Responses: We thank the Reviewer's excellent suggestions. We agree that the clarification of statistical analysis strategy is indeed crucial for illustrating the effectiveness of this inducible system.

For Fig. 5D, rather than analyzing genome editing efficiency, we aimed to determine the proportions of WT, heterozygous, or homozygous PGCs within individual embryos by genotyping single PGCs from each embryo using Sanger sequencing peak patterns, instead of relying on monoclonal sequencing (Fig. R12A). The percentage of mutated PGCs per embryo is quite important for studying the gene functions in PGC development. In addition, we conducted genome efficiency analysis in this experiment by Sanger sequencing of monoclonal with pooled PGCs by sorting, and found that the mutation rate in PGCs within 4-OHT-treated embryos was about 34.6% (9/26), while no mutations in the *tbx16* locus were detected in DMSO-treated embryos or somatic cells (Fig. R12B). For Fig. 5E, this presentation is indeed quite misleading, so we removed the ratio of mutated forms to indicate how many kinds of mutation we identified.

Figure R12. Mutation efficiency of *tbx16* alleles in PGCs. **A**, Sanger sequencing peak maps showing different mutation types of *tbx16* alleles (WT, heterozygous mutation, homozygous mutation). The shadow yellow box indicates the *tbx16* target site. **B**, Bar chart showing the allelic mutation efficiency of *tbx16* in PGCs and SCs within 4-OHT or DMSO-treated *Tg(piwill:nCas9^{ERT2}-n3U;piwill:mCherry-CAAX-n3U;U6:tbx16-gRNAs; cryaa:CFP)* embryos. No. clones, the number of monoclonal for Sanger sequencing. **C**, ISH-detected *ddx4* expression in control and *tbx16* PGC-KO embryos at 24 hpf. The box

plot showing the ratios of normal and mislocated PGCs in control and PGC-KO embryos. Ne, number of embryos.

For Fig. 5G, we employed the paired *t*-test for significance analysis because we actually compared DMSO- and 4-OHT-treated embryos from the same batches, which were a paired dataset. There are several reasons we have to compare the data in pair:

- 1, The spontaneous ectopic PGC migration rates in WT embryos vary among different batches.
- 2, The proportion of homozygote PGCs varies among our *tbx16* PGC-specific KO embryos.
- 3, It is hard to trace ectopically located PGCs and also double-check their genotypes.

Thus, for each experiment, we divided embryos from one batch into two groups for 4-OHT or DMSO treatment, which is suitable for paired *t*-test analysis.

In addition, we performed *ddx4* in situ hybridization (ISH) to detect PGC locations in 4-OHT- and DMSO-treated embryos (Fig. R12C), and the ratio of mislocated PGCs in PGC-specific KO embryos was higher than that of control embryos, supporting the function of *Tbx16* in PGC migration.

Overall, the study does not provide clear evidence of the tool's efficiency in generating mutations. The authors require to do following:

1-3. Include proper controls for Fig. 4-6.

The authors should include several positive and negative controls to accurately calculate the efficiency. The authors have the *piwil:nCas9-n3U* strain. *piwil:nCas9-n3U;sgRNA* double transgenic fish can serve as a positive control. *piwil:nCas9ERT2-n3U* without sgRNA can serve as a negative control. Vehicle treatment with *piwil:nCas9ERT2-n3U; sgRNA* double transgenic fish can serve as a negative control. Also please perform NGS-based mutation analysis instead of relying on subcloning.

Responses: We thank the Reviewer for these excellent suggestions. We have included the necessary controls for adult females (Fig. 4, for the *hwa* locus), PGC-specific knockout during early embryogenesis (Fig. 5), and *Cdx2* knockout in mouse embryos (Fig. 6).

For adult females (Fig. 4), we used DMSO-treated females as negative controls, as shown in Fig. R6. Due to the limitation of 4-OHT absorption, we supposed that, in this case, the constitutively expressed Cas9 system (*piwil:nCas9-n3U;sgRNA* double transgenic fish) should induce a higher mutation rate than our inducible system. However, our system offers a valuable advantage when the target genes are essential for early gonad development, yet we want to study their functions during oogenesis or early embryonic development. Our inducible system can achieve this goal despite relatively lower efficiency at this stage, making it a useful approach for the community.

For PGC-specific knockout during early embryogenesis (Fig. 5), first, we included DMSO treatment as a control (Fig. R12B). Meanwhile, dedicated comparisons about genome editing

between *piwil:nCas9-n3U* and *piwill:nCas9^{ERT2}-n3U* have been conducted in the *tbxta* locus, as shown in Fig. R11. Second, for the *tbx16* locus, we performed NGS of PGCs' genomes in our inducible system or in WT embryos injected with commercial Cas9 protein and gRNAs at the 1-cell stage, as a strong positive control (Fig. R2A, this figure index is consistent with the response to Reviewer 1#). The mutation rate in positive control PGCs is 57.8%, whereas the mutation rate of our system is 41.7% (Fig. R2B), which is comparable and would make little difference in application. But considering high rates of non-specific somatic cell mutation in the *piwil:nCas9-n3U* line, the relatively low but sufficient efficiency of our system does not obscure its superior PGC specificity.

Figure R2. Off-target analysis of the *tbx16* locus with WGS. **A**, Schematic showing PGC collection and WGS. **B**, Bar chart showing the mutation efficiency in *tbx16* target sites and predicted potential off-targeting sites through the whole genome, in PGCs within 4-OHT or DMSO treated *Tg(piwil1:nCas9^{ERT2}-n3U; piwil1:mCherry-CAAX-n3U; U6:tbx16-gRNAs; cryaa:CFP)* embryos (left, inducible editing system) and within Cas9 injected or WT embryos (right, exogenous Cas9 protein). Red letters indicate the sequence of *tbx16* target sites. Target, *tbx16* gRNA target sites; R1 to R9, the predicted

tbx16 gRNA off-target sites by Cas-OFFinder (Bae et al., 2014). INDEL, insertion-deletion; REF, reference genome; SNV, single-nucleotide variant.

For *Cdx2* knockout in mouse embryos (Fig. 6), we included three groups of negative control: C1, injection of nCas9^{ERT2} and *nls-mCherry* mRNA with DMSO treatment; C2, injection of nCas9^{ERT2} and *nls-mCherry* mRNA treated with 4-OHT; C3, injection of nCas9^{ERT2}, *nls-mCherry* mRNA, and *Cdx2* gRNAs with DMSO treatment. All these control embryos displayed normal morphology at the blastocyst stage with a proper ratio of CDX2-positive cells (Fig. 6E-F). For genome editing efficiency detection, we just focus on the C3 control embryos and found no mutations in the C3 embryos, compared with about a 60% mutation rate in the experimental group at the *Cdx2* locus (Fig. 6B).

1-4. Somatic cells

The authors require to use somatic or non-PGC cells to determine spatial piwil:nCas9ERT2-n3U activity for Fig. 4 and 5.

Responses: We sincerely thank the Reviewer for pointing this out. This suggestion is similar to the Reviewer 1#'s comment 3. Accordingly, we have organized the relevant data into a new Fig. R13 to facilitate your review.

For Fig. 4, we collected ovaries from *Tg(piwil:nCas9^{ERT2}-n3U;U6:hwa-gRNAs;ef1a:GFP)* young females at 45 dpf treated with 4-OHT or DMSO. Then, we performed immunostaining of Cas9 in their oocytes after 1 day post-treatment and found that the Cas9 signals were only present in the oocytes, but not in the surrounding gonadal somatic cells (Fig. R13A-B).

For Fig. 5, we performed the genotyping of PGCs and SCs of 4-OHT or DMSO-treated *Tg(piwil:nCas9^{ERT2}-n3U;piwil:mCherry-CAAX-n3U;U6:tbx16-gRNAs;cryaa:CFP)* embryos with monoclonal Sanger sequencing and found no *tbx16* mutations in SCs within either 4-OHT or DMSO-treated embryos (Fig. R13C). Furthermore, we also performed immunostaining of Tbx16 protein in 4-OHT and DMSO-treated embryos at the bud stage, and there was no significant difference in Tbx16 protein levels in SCs between 4-OHT and DMSO-treated embryos (Fig. R13D-E), which supports no loss of Tbx16 protein in somatic cells.

Figure R13. PGC-specific mutations induced by nCas9^{ERT2} in oocytes and embryos. **A**, Immunostaining of Cas9 (green) and nuclei stained by DAPI (blue) in ovaries within the 4-OHT or DMSO-treated females at 45 dpf. **B**, Box plots showing the ratio of Cas9 (red) signal intensity in the nucleus to the cytoplasm of PGCs (left), and the signal intensity ratio of Cas9 (red) to DAPI (blue) in the nucleus of PGCs (right). Nu, nucleus. Cyto, cytoplasm. No. cells, the number of oocytes. *P*-values of *t*-test (unpaired, two-sided) were also labeled. **C**, Bar chart showing genotypes of PGCs in DMSO and 4-OHT-treated groups at 24 hpf. Soma, somatic cells. No. clones, the number of monoclones for Sanger sequencing. **D**, Immunostaining results of Tbx16 (green) and Ddx4 (red) with nuclei stained by DAPI (blue) in embryos treated with 4-OHT or DMSO from the sphere stage. **E**, Box plot showing the statistic results of the ratio of Tbx16 signals to DAPI in PGCs and somatic cells in either 4-OHT or DMSO-treated embryos. *P*-values of *t*-test (unpaired, two-sided) were also labeled.

1-5. Use reporter gene to accurately examine cas9 function

A standard approach to evaluate the efficiency of genome editing is to utilize EGFP or mCherry reporter. The authors require to use an EGFP reporter line that expresses fluorescence expression in PGC and/or non-PGC cells for their testing. Injection sgRNA rather than using a sgRNA stable line would be enough as generating a stable line does not make sense for revision. The authors require to test only PGC reporter expression is lost but non-PGC reporter expression retains. The authors already generated *piwill:EGFP* line, which may serve as a PGC reporter.

Responses: We thank the Reviewer for proposing this promising approach, which we had already tested at the very beginning of our project. Unfortunately, a major limitation of this strategy is the prominent presence of maternal EGFP protein in early *Tg(piwill:EGFP)* embryos. The *piwill* promoter drives EGFP expression in oocytes, which underlies the induction of *hwa* mutation by *piwill:nCas9^{ERT2}* in oocytes. Thus, the advantage of using a

reporter line to assess editing efficiency based solely on the fluorescence, rather than sequencing the genome, does not apply in our case. Nevertheless, the reporter line approach could be valuable if other promoters capable of driving specific zygotic PGC expression were identified or if specific reporter lines were established for early embryogenesis and the ovary in adults, respectively. However, the direct detection of genome editing efficiency remains necessary, even when reporter validation is performed.

2. Optimization

The authors used 40 μ M for early embryos and 5 μ M 4-OHT for adults. To my knowledge, 40 μ M is likely higher than standard 4-OHT concentrations used in zebrafish (up to 5-10 μ M) for other studies. The authors require to provide how efficiency is changed in different 4-OHT concentrations. If the authors already tested other 4-OHT conditions, please provide those optimization data. This data should be necessary for comprehensive analysis of their technique.

Responses: We sincerely thank the Reviewer for pointing this out. We have tested different concentrations and presented the data below.

Actually, we employed two types of 4-OHT dosing selection schemes for our system, tailored to the developmental stages.

For adult treatment, we performed a spawning rate detection by 5 or 10 μ M 4-OHT treatment (Fig. R7A, this figure index is consistent with the response to Reviewer 1#), and observed a severe disruption of reproduction in the 10 μ M 4-OHT group (Fig. R7B). Thus, although the relatively lower editing efficiency, we still chose a concentration of 5 μ M for adult treatment to ensure the spawning rate.

Next, for early embryogenesis, we tested four concentrations of 4-OHT: 25, 30, 40, and 50 μ M from the sphere to the 2 dpf stages, and detected the survival rates of treated embryos at the 2 dpf, indicating a severe cytotoxic effect with the 50 μ M treatment (Fig. R7C-D). In addition, by the T7EI assay, we found that the genome editing efficiency of 40 μ M 4-OHT treatment was higher than that of 25 μ M (Fig. R7E). Via immunostaining of Cas9 protein, we revealed that 40 μ M 4-OHT treatment was sufficient to trigger nCas9^{ERT2} nuclear translocation in PGCs (Fig. R7F). Thus, a concentration of 40 μ M was used for early embryos in the subsequent experiments in our manuscript. We added all these results to the revised supplementary figure (Fig. S5).

Figure R7. 4-OHT dosing selection for embryos and adults. **A**, Schematic showing the 4-OHT treatment with 45 dpf females and detection of spawning rates. **B**, Bar chart showing the spawning rates of adults treated with 10 μM or 5 μM 4-OHT at 45 dpf. Adults: the serial number of treated transgenic zebrafish. **C**, Schematic showing the 4-OHT treatment for embryos. **D**, Table showing survival rates of embryos treated with different concentrations of 4-OHT from the sphere stage to the 2 dpf stage. **E**, Gel electrophoretic image showing T7E1 assay outcome of the *tbx16* locus. The mutant band was indicated by the arrow. NC, negative control. **F**, Immunostaining of Cas9 (red) and nuclei stained by DAPI (blue) in PGCs within the DMSO or 40 μM 4-OHT-treated embryos at 2 dpf.

Minor comments:

1. Fig. 2 labeling: Adding time points, such as 1,2,4, and 6 hpt (hours post-treatment) would be more informative than 30% Epi, Shield, 75% Epi, and Bud.

Responses: We thank the Reviewer for the suggestion. We have added the time points in the revised Fig. 2.

2. Line 161 ea:EGFP -> may typo, it could be efla:EGFP

Responses: We thank the Reviewer for the suggestion and **apologize for the typo**. We have corrected it and carefully reviewed the manuscript and figures to address all typos.

References

- Bae, S., J. Park, and J.-S. Kim. 2014. Cas-OFFinder: a fast and versatile algorithm that searches for potential off-target sites of Cas9 RNA-guided endonucleases. *Bioinformatics*. 30:1473–1475. doi:10.1093/bioinformatics/btu048.
- Schulte-Merker, S., R.K. Ho, B.G. Herrmann, and C. Nüsslein-Volhard. 1992. The protein product of the zebrafish homologue of the mouse T gene is expressed in nuclei of the germ ring and the notochord of the early embryo. *Development*. 116:1021–1032. doi:10.1242/DEV.116.4.1021.
- Wu, X., W. Shen, B. Zhang, and A. Meng. 2018. The genetic program of oocytes can be modified in vivo in the zebrafish ovary. *J Mol Cell Biol*. 10:479–493. doi:10.1093/jmcb/mjy044.

November 3, 2025

Re: JCB manuscript #202412216R

Xiaotong Wu
Tsinghua University

Dear Dr. Wu,

Thank you for submitting your revised manuscript entitled "Stage- and tissue-specific gene editing using 4-OHT inducible Cas9 in whole organism." The manuscript has been seen by the original reviewers whose full comments are appended below. While the reviewers continue to be overall positive about the work in terms of its suitability for JCB, some important issues remain.

You will see that there are several concerns about the reproducibility and editing efficiency of the method as well as inconsistencies between your data and published numbers of PGCs and oocytes. We do not think that testing additional targets is necessary but it is important to address these concerns about the targets shown in the paper.

Our general policy is that papers are considered through only one revision cycle; however, given that the suggested changes are relatively minor we are open to one additional short round of revision. Please note that we will expect to make a final decision without additional reviewer input upon resubmission.

Please submit the final revision within one month, along with a cover letter that includes a point by point response to the remaining reviewer comments.

Thank you for this interesting contribution to Journal of Cell Biology. You can contact me or the scientific editor listed below at the journal office with any questions at cellbio@rockefeller.edu.

Sincerely,

Anna Huttenlocher, MD
Monitoring Editor
Journal of Cell Biology

Dan Simon, PhD
Scientific Editor
Journal of Cell Biology

Reviewer #1 (Comments to the Authors (Required)):

The manuscript by Li et al. describes and characterizes a new inducible system for conditional gene knockout, leveraging tissue-specific expression of a Cas9-ERT2 fusion protein. After testing various construct designs, the authors demonstrate that one in particular, nCas9ERT2, shows robust nuclear localization upon 4-OHT treatment and is capable of editing target genes in HEK293T in vitro experiments and in zebrafish embryos, where primordial germ cell (PGC) fates were examined. The authors provide solid evidence for the potential of this system to complement alternative approaches, and this manuscript is likely to be of broad interest to readers in disease modeling.

The authors have adequately addressed all concerns raised previously, and the current manuscript is acceptable for publication in JCB. No further changes are recommended.

Reviewer #2 (Comments to the Authors (Required)):

I appreciate the authors' effort to address my previous comments. The revised manuscript includes new analyses and additional data; however, the main concerns raised in my initial review remain insufficiently addressed.

This study is the first technical paper to examine Cas9 fused with an estrogen receptor (ER) for conditional knockout in animal models. Therefore, it is critical that the manuscript clearly demonstrates that this technique can efficiently and effectively induce conditional knockout in the target cells - specifically, primordial germ cells (PGCs) in this study.

The authors tested three zebrafish genes: *tbxta*, *hwa*, and *tbx16*. The authors used *tbxta* as a negative control to verify that no mutations occurred in PGCs, as *tbxta* knockout in these cells does not result in observable defects. The editing efficiency for *hwa* is extremely low and highly variable among individuals, with mutation rates ranging from 1.71% to 14.8%. Such low and inconsistent mutation rates are insufficient for functional analyses of other genes, limiting the potential for broader application by other researchers. For *tbx16*, 9 out of 26 animals (34.6%) displayed mutations following 4-OHT treatment, and Figure 5D indicates a biallelic mutation rate of 23.8% (5 out of 21). These mutation rates are also low. Moreover, the reported PGC migration phenotype is unclear, likely due to technical limitations described in the authors' response. If technical issues prevent reliable phenotype assessment, *tbx16* may not be an appropriate target gene to evaluate this genetic tool. To generate reliable and reproducible data, the authors should test additional targets - such as cell cycle-related genes, apoptosis regulators, or GFP reporter genes - that would yield more definitive phenotypic outcomes.

The authors acknowledge variability and the need for optimization (lines 215-217): "It is worth noting that the ratio of ventralized embryos was low and varied among individuals." They also note that "Further optimization of 4-OHT treatments may be needed to achieve a higher gene editing efficiency." However, because the Cas9-ER concept has already been introduced in previous publications (Oakes et al., *Nat Biotechnol* 2016; Liu et al., *Nat Chem Biol* 2016; Zhao et al., *Mol Ther Nucleic Acids* 2018; Lu et al., *Nucleic Acids Res* 2018), such optimization should have been completed within this study. Both the original and revised versions fail to provide clear validation of efficiency or experimental conditions sufficient for reproducible results.

The main reason for the low efficiency is insufficient nCas9ERT2 expression in PGCs. In response to my previous question - "How many Cas9⁺ PGCs are detectable per embryo and in the gonad?" - the authors added new data in Fig. 1E and 4C. In Fig. 1E, the number shown above the graph presumably represents total PGCs analyzed for subcellular Cas9 localization among *ddx4*⁺ PGCs. I would like to note that published data for the PGC number in zebrafish (Tzung et al., *Stem Cell Reports* 2015; Ye et al., *Mar Biotechnol (NY)* 2019) indicate 25 - 44 and 12 - 70 PGCs at 24 hpf, respectively. These are substantially lower than the 124 reported here. In this case, the authors do not provide Cas9⁺ cell number among total PGCs in the individual larva. If 124 represents average PGCs per embryo, this conflicts with Fig. 2, which shows only one or two PGCs across all examined time points.

In Figure 4C, ovary images from adult fish display only two or three Cas9⁺ cells. A typical zebrafish ovary contains many more oocytes (see *Biol Reprod* 2023 109(5):586-600; *Nat Commun* 2024 15:5248; *Cell* 2007 129:69-82). Moreover, no quantification is provided, making it impossible to determine how many oocytes express Cas9 in *piwil1:nCas9ERT2-n3U*. Thus, Cas9⁺ cell counts per larval PGC and per adult gonad remain unquantified, making it difficult to determine the cause of the low efficiency.

In summary, the authors have developed a potentially valuable genetic tool for conditional knockout; however, the current data show insufficient editing efficiency to support broad claims of general applicability.

清华大学

COLLEGE OF LIFE SCIENCES, TSINGHUA UNIVERSITY

Beijing 100084, China. Tel: (10) 62772256, Fax: (10) 62794401. Email: wuxt@mail.tsinghua.edu.cn

November 14, 2025

Dan Simon, PhD.
Scientific Editor, *Journal of Cell Biology*,

Dear Dan:

We sincerely appreciate for giving us the opportunity to further respond and clarify our work titled “*Stage- and tissue-specific gene editing using 4-OHT inducible Cas9 in whole organism*”. Regarding the remaining concerns raised by Reviewer #2, we have revised our manuscript and prepared the following response.

Generally, the remaining concerns focus on the efficiency and reproducibility of our system. We would like to clarify the following points:

1) For the relatively low editing efficiency, we faithfully presented the mutation efficiencies of our method across different target sites, different stages, and treatment conditions in the manuscript. Notably, although the efficiencies varied, all of them exhibited clear evidence of successful editing. In particular, we further added the NGS-based validation, confirming it in our revised manuscript. **Therefore, this approach achieves targeted genome editing in PGCs and oocytes in adult ovaries and allows us to interrogate maternal gene function and to distinguish gene roles between somatic cells and PGCs in early embryos, which are not readily or easily attainable with existing methods.**

2) Regarding testing additional genes, we believe that screening a large number of genes would eventually identify some with higher editing efficiency; however, such screening is not necessary for demonstrating the feasibility of a new method. Furthermore, only reporting the most efficient cases would not be an objective representation of the tool’s overall performance and would lead to biased conclusions.

3) For the expression of nCas9^{ERT2} in PGCs, our data show that **more than 75% PGCs displayed Cas9 expression driven by the *piwill* promoter we optimized** (as shown in Fig. 1E). Additionally, here we showed the statistical tables for individual embryos at each stage we used, and the values are not affected by whether the individual embryo statistic or the combined statistic is used. Furthermore, this promoter was able to drive Cas9 expression in PGCs efficiently and specifically, which leads to efficient target gene editing in our another project. If you would like to read it, we have already posted it as a preprint on bioRxiv (<https://doi.org/10.1101/2025.11.04.686672>) (under revision elsewhere).

POINT-BY-POINT RESPONSE

清华大学

COLLEGE OF LIFE SCIENCES, TSINGHUA UNIVERSITY

Beijing 100084, China. Tel: (10) 62772256, Fax: (10) 62794401. Email: wuxt@mail.tsinghua.edu.cn

Reviewer: 2

The authors tested three zebrafish genes: *tbxta*, *hwa*, and *tbx16*. The authors used *tbxta* as a negative control to verify that no mutations occurred in PGCs, as *tbxta* knockout in these cells does not result in observable defects. The editing efficiency for *hwa* is extremely low and highly variable among individuals, with mutation rates ranging from 1.71% to 14.8%. Such low and inconsistent mutation rates are insufficient for functional analyses of other genes, limiting the potential for broader application by other researchers. For *tbx16*, 9 out of 26 animals (34.6%) displayed mutations following 4-OHT treatment, and Figure 5D indicates a biallelic mutation rate of 23.8% (5 out of 21). These mutation rates are also low. Moreover, the reported PGC migration phenotype is unclear, likely due to technical limitations described in the authors' response. If technical issues prevent reliable phenotype assessment, *tbx16* may not be an appropriate target gene to evaluate this genetic tool. To generate reliable and reproducible data, the authors should test additional targets - such as cell cycle-related genes, apoptosis regulators, or GFP reporter genes - that would yield more definitive phenotypic outcomes.

Responses: Each gene we tested in our paper reflects a distinct scientific question that our method is designed to address.

1, *tbxta* locus. The *tbxta* locus is not a “negative control”, but a somatic cell control. “*tbxta* is zygotically expressed in mesendodermal progenitors and the notochord.” (Lines 151-152 in the revised manuscript). Loss of *tbxta* in early embryos causes truncation of the posterior trunk and lack of notochord, primarily due to *tbxta* deficiency in mesendodermal progenitors and notochord cells (somatic cells, but not PGCs). **Therefore, the *tbxta* locus provides a sensitive readout for unintended leaky genome editing in somatic cells in our system.** Our results indeed demonstrated “an efficient mutation rate (37.1%) of the *tbxta* locus induced by nCas9^{ERT2} in PGCs.....no mutated reads in somatic cells.....” (Lines 177-181) and “embryos showed a normal notochord morphology” (Lines 166-167), indicating PGC specificity in our system.

Thus, it is not “no mutations occurred in PGCs”, but there are efficient mutations in PGCs and no mutations in somatic cells, which supports the cell-specificity of our genome editing system.

2, *hwa* locus. The *hwa* gene is a typical example of maternal genes, which means that its expression is very important to either oogenesis or early embryonic development.

Due to the lack of efficient strategies for knocking out maternal genes, current approaches to studying maternal gene function in zebrafish generally fall into two categories. 1) For maternal genes whose loss does not affect zygotic development, researchers typically generate homozygous mutant females by raising at least 4 generations, which requires approximately 12 months. 2) For maternal genes whose absence causes embryonic lethality or female infertility, making it impossible to obtain homozygous mutant females, the only feasible strategy is to knock out or knock down the maternal gene during oocyte maturation. **At present, highly efficient and user-friendly methods for maternal gene KO or KD are still challenging.** In this study, we tested the *hwa* locus to demonstrate that our system enables the acquisition of maternal mutant phenotypes in F0 females. Although the efficiency is relatively low, classical ventralized phenotypes can be rapidly and reliably obtained by screening mosaic females with germline mutations. Therefore, achieving about 10% average

清华大学

COLLEGE OF LIFE SCIENCES, TSINGHUA UNIVERSITY

Beijing 100084, China. Tel: (10) 62772256, Fax: (10) 62794401. Email: wuxt@mail.tsinghua.edu.cn

efficiency in maternal gene knockout is already sufficient to support several functional analyses of maternal genes in some cases, which offers researchers a new avenue to solve long-standing problems in maternal-effect gene research.

3, *tbx16* locus. Tbx16 has also been reported to participate in regulating the migration of PGCs. In 24 hpf *tbx16* zygotic mutant (Ztbx16) embryos, a portion of PGCs is ectopically localized in the head or tail region of embryos (Weidinger et al., 1999). The authors speculated that this phenotype is caused by a deficiency of Tbx16 protein, which causes an aberrant expression pattern of *cxcl12a* in somatic cells and disrupts PGC migration. **Thus, we tested whether specifically KD *tbx16* in PGCs could affect PGC migration, and demonstrated that, even with 34.6% mutation rates, loss of Tbx16 in PGCs leads to PGC mislocations in the head.**

The authors acknowledge variability and the need for optimization (lines 215-217): "It is worth noting that the ratio of ventralized embryos was low and varied among individuals." They also note that "Further optimization of 4-OHT treatments may be needed to achieve a higher gene editing efficiency." However, because the Cas9-ER concept has already been introduced in previous publications (Oakes et al., Nat Biotechnol 2016; Liu et al., Nat Chem Biol 2016; Zhao et al., Mol Ther Nucleic Acids 2018; Lu et al., Nucleic Acids Res 2018), such optimization should have been completed within this study. Both the original and revised versions fail to provide clear validation of efficiency or experimental conditions sufficient for reproducible results.

Responses: In our previous response letter, we clarified that our system optimized PGC-specific promoter (**achieving tissue-specific**) and 4-OHT treatment in early embryos and adult female fish (**achieving stage-specific**), which can not be achieved by established Cas9-ER-based methods (only tested in cell lines). The following are the detailed statements in our previous response letter:

"Specifically, to achieve spatiotemporal gene editing, we performed the following optimizations:

1, Through immunostaining of the Cas9 protein, we screened various combinations of Cas9, NLS, and ERT2, and identified that the nCas9^{ERT2} protein flanked by two different nuclear localization signals on either side would efficiently improve its translocation into the nucleus upon induction of 4-OHT (Fig. S1 and 1A). In addition, we validated its genome editing efficiency by T7EI assay (Fig. 1B).

2, Optimization of the germ-line specific promoter. Besides optimization of the *piwill* promoter, which has been shown in Fig. S2, we also tested *ca15b* and *tdrd7a* promoters by constructing transgenic lines and checked the reporter fluorescence protein expression. However, the expression of mCherry driven by these promoters displayed strong maternal expression, a ubiquitous expression pattern at early stages, and relatively weak PGC-specific expression till at the bud stage. Thus, these two promoters were not suitable for PGC editing during the early stages, and we did not include these results in our previous figures, nor do we add them in the revised version.

3, We employed two types of 4-OHT dosing selection schemes for our system, tailored to the developmental stages. For early embryogenesis, we tested four concentrations of 4-OHT from the sphere to the 2 dpf stages, and checked the survival rates, nCas9^{ERT2} nuclear translocation, and editing

清华大学

COLLEGE OF LIFE SCIENCES, TSINGHUA UNIVERSITY

Beijing 100084, China. Tel: (10) 62772256, Fax: (10) 62794401. Email: wuxt@mail.tsinghua.edu.cn

rates of treated embryos. For adult treatment, we performed a spawning rate detection using two concentrations of 4-OHT treatment. Then, we chose the 4-OHT concentrations, ensuring low cell toxicity and higher editing efficiency.”

The main reason for the low efficiency is insufficient nCas9ERT2 expression in PGCs. In response to my previous question - "How many Cas9+ PGCs are detectable per embryo and in the gonad?" - the authors added new data in Fig. 1E and 4C. In Fig. 1E, the number shown above the graph presumably represents total PGCs analyzed for subcellular Cas9 localization among ddx4+ PGCs. I would like to note that published data for the PGC number in zebrafish (Tzung et al., Stem Cell Reports 2015; Ye et al., Mar Biotechnol (NY) 2019) indicate 25 - 44 and 12 - 70 PGCs at 24 hpf, respectively. These are substantially lower than the 124 reported here. In this case, the authors do not provide Cas9+ cell number among total PGCs in the individual larva. If 124 represents average PGCs per embryo, this conflicts with Fig. 2, which shows only one or two PGCs across all examined time points.

Responses: Based on our data, we do not agree that the relatively low efficiency is primarily due to insufficient nCas9ERT2 expression in PGCs. For Fig. 1E, we showed the number of total PGCs we analyzed from multiple embryos at different stages without 4-OHT treatment. Here, we also showed raw statistical results in tables (as shown below). The ratio of Cas9-positive PGCs in individual embryos was higher than 75% on average at various stages. The ratios of Cas9-positive PGCs are not affected by whether the individual embryo statistic or the combined statistic is used.

Table. The numbers of Ddx4+ and Ddx4+; Cas9+ double positive PGCs in individual embryos at different stages

Sphere			30% epi			Shield			75% epi			Bud			24 hpf		
Ddx4+	Ddx4+; Cas9+	Ratio	Ddx4+	Ddx4+; Cas9+	Ratio	Ddx4+	Ddx4+; Cas9+	Ratio	Ddx4+	Ddx4+; Cas9+	Ratio	Ddx4+	Ddx4+; Cas9+	Ratio	Ddx4+	Ddx4+; Cas9+	Ratio
1	1	1.00	1	1	1.00	1	1	1.00	2	2	1.00	2	2	1.00	19	19	1.00
1	1	1.00	2	2	1.00	2	0	0.00	8	7	0.88	2	2	1.00	16	16	1.00
2	2	1.00	3	3	1.00	6	6	1.00	3	3	1.00	16	14	0.88	9	9	1.00
1	1	1.00	5	4	0.80	4	3	0.75	8	7	0.88	4	4	1.00	14	14	1.00
			2	1	0.50	1	1	1.00	4	2	0.50	2	2	1.00	15	15	1.00
			2	2	1.00	4	1	0.25	4	4	1.00	4	2	0.50	9	9	1.00
			2	0	0.00	6	4	0.67	1	1	1.00	2	2	1.00	11	11	1.00
			4	4	1.00	4	4	1.00	4	4	1.00	3	3	1.00	20	20	1.00
			3	3	1.00	1	1	1.00	1	1	1.00	1	1	1.00	23	23	1.00
			1	1	1.00	5	2	0.40	5	2	0.40	8	7	0.88	9	9	1.00
			1	1	1.00	1	1	1.00	5	4	0.80				7	7	1.00
			1	1	1.00	2	0	0.00	4	4	1.00				10	10	1.00
			5	4	0.80	5	4	0.80							11	11	1.00
			6	6	1.00	6	6	1.00							5	5	1.00
			1	1	1.00	1	1	1.00							11	11	1.00
			1	1	1.00	1	1	1.00							7	7	1.00
			5	5	1.00	3	3	1.00							4	4	1.00
			3	3	1.00	2	2	1.00									
			2	2	1.00	4	2	0.50									
			4	2	0.50	1	1	1.00									
			1	1	1.00												

At early stages, such as the sphere, PGCs were located as four highly dispersed clusters in the margin region of embryos. Thus, we can not change the position of fixed and mounted embryos under the confocal microscope to take pictures of every cluster of PGCs. The number of PGCs at that stage was relatively lower. However, at later stages, almost every PGCs expressed Cas9.

In Fig. 2, we detected the subcellular localization of Cas9 in DMSO and 4-OHT-treated embryos, which is not the same experiment as shown in Fig. 1E. And there were no 24 hpf embryos in the assay in Fig. 2.

In Figure 4C, ovary images from adult fish display only two or three Cas9+ cells. A typical zebrafish ovary contains many more oocytes (see Biol Reprod 2023 109(5):586-600; Nat Commun 2024

清华大学

COLLEGE OF LIFE SCIENCES, TSINGHUA UNIVERSITY

Beijing 100084, China. Tel: (10) 62772256, Fax: (10) 62794401. Email: wuxt@mail.tsinghua.edu.cn

15:5248; Cell 2007 129:69-82). Moreover, no quantification is provided, making it impossible to determine how many oocytes express Cas9 in piwil1:nCas9ERT2-n3U. Thus, Cas9⁺ cell counts per larval PGC and per adult gonad remain unquantified, making it difficult to determine the cause of the low efficiency.

Responses: To detect the Cas9 protein expression in the ovary, we collected oocytes from 8 DMSO- and 8 4-OHT-treated females. Then we mixed oocytes from different females to perform Cas9 staining assay, and the statistical results were shown in Fig. 4D. Maybe the labels “No. cells” are a little bit misleading, so we replaced them with “No. oocytes” in the revised figure (as shown here).

Meanwhile, we added information about zebrafish oocyte immunostaining and data availability in the Methods section. And we further revised our manuscript accordingly and highlighted modifications, including some minor changes in figures related to labeling.

Thank you very much for your time and consideration, and I look forward to hearing from you and the reviewers!

Yours sincerely,

Xiaotong

Xiaotong Wu, Ph.D.
Assistant Researcher
School of Life Sciences, Medical Science Bldg C126
Tsinghua University, Haidian District
Beijing, China, 100084
Office: 86-10-62772435
Email: wuxt@mail.tsinghua.edu.cn

December 1, 2025

RE: JCB Manuscript #202412216RR

Xiaotong Wu
Tsinghua University

Dear Dr. Wu,

Thank you for submitting your revised manuscript entitled "Stage- and tissue-specific gene editing using 4-OHT inducible Cas9 in whole organism." We would be happy to publish your paper in JCB pending final revisions necessary to meet our formatting guidelines (see details below).

A. MANUSCRIPT ORGANIZATION AND FORMATTING:

1) Text limits: Character count for Tools is < 40,000, not including spaces. Count includes title page, abstract, introduction, results, discussion, and acknowledgments. Count does not include materials and methods, figure legends, references, tables, or supplemental legends.

2) Figure formatting: Tools may have up to 10 main text figures. Scale bars must be present on all microscopy images, including inset magnifications. Please add a scale bar for the inset magnifications in figure 1C.

Also, please avoid pairing red and green for images and graphs to ensure legibility for color-blind readers. If red and green are paired for images, please ensure that the particular red and green hues used in micrographs are distinctive with any of the colorblind types. If not, please modify colors accordingly or provide separate images of the individual channels.

3) Statistical analysis: Error bars on graphic representations of numerical data must be clearly described in the figure legend. The number of independent data points (n) represented in a graph must be indicated in the legend. Please indicate whether 'n' refers to technical or biological replicates (i.e. number of analyzed cells, samples or animals, number of independent experiments). If independent experiments with multiple biological replicates have been performed, we recommend using distribution-reproducibility SuperPlots (please see Lord et al., JCB 2020) to better display the distribution of the entire dataset, and report statistics (such as means, error bars, and P values) that address the reproducibility of the findings.

Statistical methods should be explained in full in the materials and methods. For figures presenting pooled data the statistical measure should be defined in the figure legends. Please also be sure to indicate the statistical tests used in each of your experiments (both in the figure legend itself and in a separate methods section) as well as the parameters of the test (for example, if you ran a t-test, please indicate if it was one- or two-sided, etc.). Also, if you used parametric tests, please indicate if the data distribution was tested for normality (and if so, how). If not, you must state something to the effect that "Data distribution was assumed to be normal but this was not formally tested."

4) Materials and methods: Should be comprehensive and not simply reference a previous publication for details on how an experiment was performed. Please provide full descriptions (at least in brief) in the text for readers who may not have access to referenced manuscripts. The text should not refer to methods "...as previously described."

5) For all cell lines, vectors, strains, constructs/cDNAs, etc. - all genetic material: please include database / vendor ID (e.g. Addgene, ATCC, etc.) or if unavailable, please briefly describe their basic genetic features, even if described in other published work or gifted to you by other investigators (and provide references where appropriate). Please be sure to provide the sequences for all of your oligos: primers, si/shRNA, RNAi, gRNAs, etc. in the materials and methods. You must also indicate in the methods the source, species, and catalog numbers/vendor identifiers (where appropriate) for all of your antibodies, including secondary. If antibodies are not commercial, please add a reference citation if possible.

6) Microscope image acquisition: The following information must be provided about the acquisition and processing of images:

- a. Make and model of microscope
- b. Type, magnification, and numerical aperture of the objective lenses
- c. Temperature
- d. Imaging medium
- e. Fluorochromes

f. Camera make and model

g. Acquisition software

h. Any software used for image processing subsequent to data acquisition. Please include details and types of operations involved (e.g., type of deconvolution, 3D reconstitutions, surface or volume rendering, gamma adjustments, etc.).

7) References: There is no limit to the number of references cited in a manuscript. References should be cited parenthetically in the text by author and year of publication. Abbreviate the names of journals according to PubMed.

8) Supplemental materials: Tools may have up to 5 supplemental figures and 10 videos. Please also note that tables, like figures, should be provided as individual, editable files. A summary of all supplemental material should appear at the end of the Materials and methods section. Please include one brief sentence per item.

9) eTOC summary: A ~40-50 word summary that describes the context and significance of the findings for a general readership should be included on the title page. The statement should be written in the present tense and refer to the work in the third person. It should begin with "First author name(s) et al..." to match our preferred style.

10) Conflict of interest statement: JCB requires inclusion of a statement in the acknowledgements regarding competing financial interests. If no competing financial interests exist, please include the following statement: "The authors declare no competing financial interests." If competing interests are declared, please follow your statement of these competing interests with the following statement: "The authors declare no further competing financial interests."

11) A separate author contribution section is required following the Acknowledgments in all research manuscripts. All authors should be mentioned and designated by their first and middle initials and full surnames. We encourage use of the CRediT nomenclature (<https://casrai.org/credit/>).

12) ORCID IDs: ORCID IDs are unique identifiers allowing researchers to create a record of their various scholarly contributions in a single place. Please note that ORCID IDs are required for all authors. At resubmission of your final files, please be sure to provide your ORCID ID and those of all co-authors.

13) Journal of Cell Biology now requires a data availability statement for all research article submissions. These statements will be published in the article directly above the Acknowledgments. The statement should address all data underlying the research presented in the manuscript. Please visit the JCB instructions for authors for guidelines and examples of statements at (<https://rupress.org/jcb/pages/editorial-policies#data-availability-statement>).

B. FINAL FILES:

Thank you for your attention to these final processing requirements. Please revise and format the manuscript and upload materials within 7 days. If you need an extension for whatever reason, please let us know and we can work with you to

determine a suitable revision period.

Thank you for this interesting contribution, we look forward to publishing your paper in Journal of Cell Biology.

Sincerely,

Anna Huttenlocher, MD
Monitoring Editor
Journal of Cell Biology

Dan Simon, PhD
Scientific Editor
Journal of Cell Biology